# Whose Pictures, Whose Reality? Lines of Tradition in the Development of Topics, Negativity, and Power in the Photojournalistic Competition World Press Photo

Alexander Godulla *, Daniel Seibert and Rosanna Planer

Institute for Communication and Media Studies, Leipzig University, 04109 Leipzig, Germany;
daniel.seibert@uni-leipzig.de (D.S.); rosanna.planer@uni-leipzig.de (R.P.)
* Correspondence: alexander.godulla@uni-leipzig.de

**Abstract:** Initially founded in 1955 as a platform for Dutch photojournalists to increase international exposure, the World Press Photo competition has grown into the most prestigious contest of photojournalism worldwide, making it an important arena for journalism research. Using qualitative and quantitative content analyses, this study examines all photos shown in the competitions from 1960 to 2020 ($N = 11,789$) considering the origin of jury members ($N = 686$), participants ($N = 132,800$), placements ($N = 2347$) and the Human Development Index (HDI) of the countries. The topics displayed on the photos, their degree of negativity, and potential power structures in the photos are analysed over time both in terms of continental and HDI-related differences. Significant results show that Africa, Asia, and South America are more frequently depicted by the topic conflict and characterised by negative images than continents with industrialised nations (Australia/Oceania, Europe, North America). Participating European countries have a significantly higher average number of jury members, participants, and placements than participating countries from Africa, Asia, and South America, which seems to account for a dominant Eurocentric view. Implications and critical discussions are summarized in three interim conclusions at the end of this extended paper.

**Keywords:** World Press Photo; photojournalism; news photography; contest; ethics; news values; Human Development Index

## 1. Introduction

In 2021, the world of press photography experienced a small revolution. Announced on April 15, the World Press Photo of the Year featured overtly happy people for the first time in the history of the prestigious competition. In the photo, 85-year-old Rosa Luzia Lunardi is being embraced intimately by nurse Adriana Silva da Costa Souza. The moment was captured by the Danish photojournalist Mads Nissen during the COVID-19 pandemic in Brazil. He sums up his photo, titled "The First Embrace", as follows: "To me, this is a story about hope and love in the most difficult times" (World Press Photo 2021a). Jury member Kevin WY Lee adds: "I read vulnerability, loved ones, loss and separation, demise, but, importantly, also survival—all rolled into one graphic image. If you look at the image long enough, you'll see wings: a symbol of flight and hope" (World Press Photo 2021a).

Why is this remarkable? First, it should be noted that the prize, which has been awarded annually since 1955 by the Dutch foundation World Press Photo, was originally created only so that photojournalists based in the Netherlands could gain exposure to an international audience. Over the decades, its reach and global attention have increased significantly. On its 50th anniversary, the foundation could therefore confidently state that it is in a position "where it not only runs the world's most prestigious contest of photojournalism, but administers the world's widest-ranging annual photo exhibition, and offers a breadth of related activities that is unmatched" (World Press Photo 2005, p. 3).

The winning image in the Photo of the Year competition is therefore significant for the worldwide development of professional photojournalism per se, and in 2021 this differs radically from its 62 predecessors in that it presents a globally significant event from an exclusively positive perspective. In fact, this has never been the case before. To illustrate this, Argentine photographer Diego Goldberg, as chairman of the 2005 jury, wrote: "There's blood every year and to many this is disturbing, as it should be" (Goldberg 2005, p. 9).

Negativism, in other words, is usually the dominant standard in the selection of the winning photograph. Accordingly, it also characterises three especially well-known World Press Photos of the Year. They show the street shooting of a Viet Cong member by a South Vietnamese police chief in 1968 (World Press Photo 2021b), the escape of a naked Vietnamese girl from a napalm attack on the village of Trang Bang in 1972 (World Press Photo 2021c), and a lone student standing in the way of a line of tanks in Tiananmen Square in 1989 (World Press Photo 2021d). The first two examples are so well known that German artist Michael Schirner Zang included them in his exhibition "Pictures in our minds" in 1985, where visitors saw only large black canvases on which famous photographs were succinctly described in white lettering (Schirner 2017). The images were so deeply anchored in the collective memory that this was enough to reconstruct them in visitors' minds.

This extended paper is devoted to the World Press Photo phenomenon from a communication studies perspective. It interprets the photographs given awards by the foundation and presented in worldwide exhibitions as a highly selective storehouse of images that has grown over decades and through whose analysis essential lines of tradition in global photojournalism can be identified. It engages equally with the overall winning photographs, the exhibitions around the world, and the juries that select a few hundred photographs from tens of thousands of submissions every year. In this way, it aims to establish that these photos, in terms of their choice of subjects, their technical design and artistic quality, as well as the relevance of the events depicted, meet the highest standards set by professional photojournalism over time.

To provide a theoretical framework for this project, this paper first shows the factors that influence photojournalism and the special role played by the World Press Photo competition. Subsequently, the existing literature on the topic of World Press Photo is systematised and explained in a differentiated manner. The research questions derived from this serve as the basis for a qualitative content analysis of all World Press Photos of the Year identified between 1955 and 2020 ($N = 62$). With the help of hypotheses derived from the qualitative analysis, all photos shown in the competitions up to 2020 are analysed ($N = 11,789$). In addition, an analysis of the national and continental origin of all previous jury members ($N = 686$) and participants ($N = 132,800$) is carried out and placed in relation to all successful competition entries up to 2020 ($N = 2347$) as well as the development status of the respective countries. Overall, this shows how aspects such as genres, topics, negativity and power have developed within the competition. Finally, this is taken as a starting point to formulate a general assessment of the development of photojournalism from the post-war period to the present.

## 2. The Institution of Photojournalism and the Role of World Press Photo

To contextualize the topic at hand, we first consider the interdependent relationship between (1) the field of photojournalism, (2) the professional actors operating according to its rules and standards, (3) the World Press Photo Contest and its (4) dissemination, and (5) public communication itself will be examined. Each of the five strands mentioned will be briefly delimited by means of central aspects. Figure 1 shows the relationship between these topics and how they interact with each other.

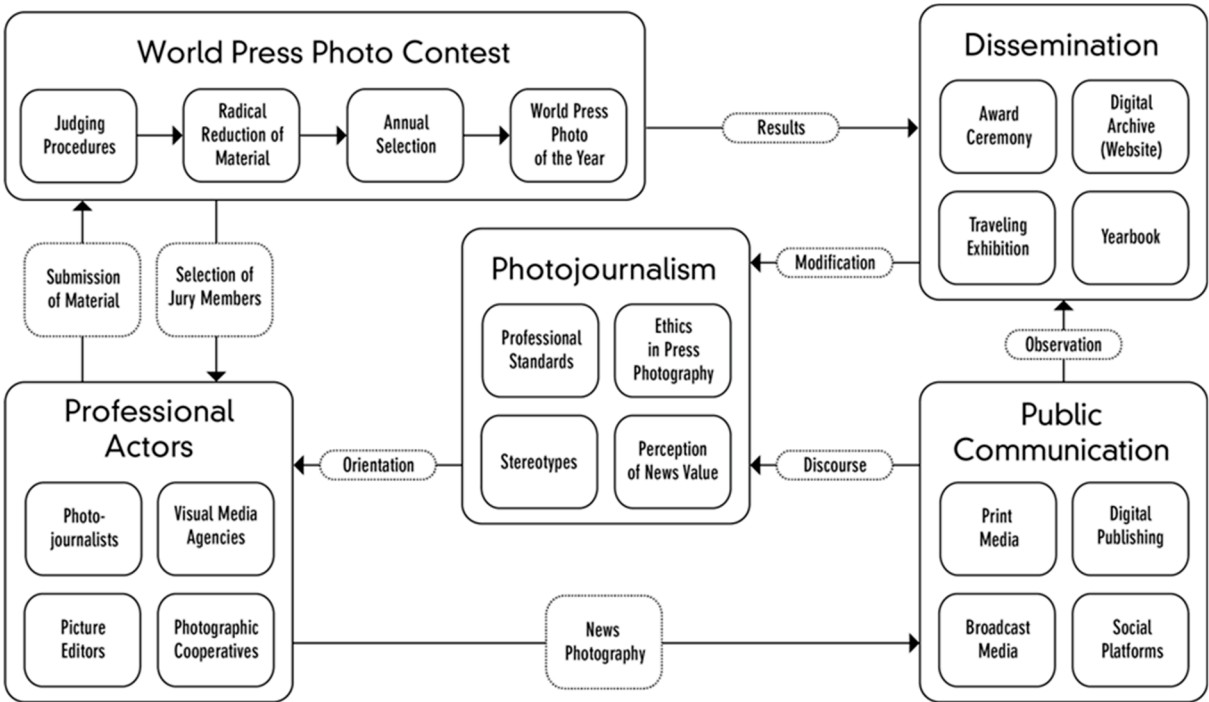

**Figure 1.** The influence of World Press Photo on institutionalised photojournalism.

*2.1. Photojournalism*

At the centre of the topic under discussion lies photojournalism itself. When defining and describing photojournalism, constructs from both professional photography and professional journalism are intertwined. Hence, "(...) professional photojournalism is intrinsically defined by the coexistence of two interrelated cultural dimensions, i.e., journalistic objectivity and photographic aesthetics" (Solaroli 2015, p. 520). Both journalistic objectivity and photographic aesthetics call for professional standards to guide photojournalists through their work (Godulla 2014). Furthermore, as every person who owns a smartphone can nowadays capture newsworthy events and distribute them, these standards are highly relevant in order to secure the photojournalists' high-quality work.

2.1.1. Professional Standards in Photojournalism

Alongside traditional journalistic standards, such as aiming for objectivity, distinguishing between facts and opinion-based comments, and informing the people from the standpoint of the Fourth Estate strengthening the democracy of a country (Caple 2014, p. 356; Deuze 2004, p. 5), further professional standards explicitly related to photojournalism have emerged, for example due to digitalization and new multimedia formats where the work of photojournalists is increasingly embedded.

Besides practices of multiskilling (Planer and Godulla 2020, p. 12), where journalists "capture vision, sound and words for a number of platforms" (Caple 2014, p. 357), and which can be seen critically as a quality-reducing process of de-skilling (Caple 2014, p. 357), photojournalists also have to "fulfil their central role of making sense of the plethora of information that surrounds us" (Caple 2014, p. 357). Both aspects—the requirement to embody multiple skills as well as the need to master large amounts of information and data—influence professional visual practices and result in an increasing demand for photojournalists to tell a (digital) story with their photos. This is also apparent in photojournalism competitions: "By elevating the storytelling aspects of an image over all else and speaking to how strong photos move public opinion, the judges reinforce and legitimize how photojournalism is distinct from photography. In this way, they call upon journalism's focus on storytelling to carve out a unique space within photography" (Lough 2021, p. 317).

Further standards that help photojournalists maintain their professional integrity and thereby differentiate them from citizen photojournalists are a high degree of sensitivity, "telling a different side of a story", "inspiring others and invoking change", as well as "memory-keeping", as revealed in a study by Mortensen and Gade (2018, p. 582). Furthermore, when judging photos in photojournalism contests, it has been noted that "moments and emotion surpassed all other elements" (Lough 2021, p. 316). These qualities also become highly apparent in the context of World Press Photo, where submissions often touch sensitive topics, tell stories in themselves, and are used in order to either reminisce about a certain event or raise awareness about an issue. This also represents a historical root of modern photojournalism: As early as the 19th century, photographers began to explore the negative consequences of the industrial revolution through topics such as child labor or mass poverty. This so-called social documentary photography interprets photography as "an instrument for social reform and political action—a way to reveal the struggles, strife, and strength of individuals on the fringes of society" (Young 2008, p. 254). It focuses in particular on the authenticity of the image.

### 2.1.2. Ethics in Photojournalism

In aiming to fulfil these practices, it is crucial for photojournalists and press photographers to follow ethical standards. A key point in this discussion and critique is the practice of post-editing photos, and the fine line between "legitimate post-production" and "illegitimate manipulation" (Solaroli 2015, p. 521). After all, it led the jury of the World Press Photo Contest in 2015 to disqualify 20 percent of the submitted photos because of "excessive" post-editing (Solaroli 2015, p. 521). The veracity of photos therefore "comes to be increasingly undermined" (Solaroli 2015, p. 519), which in turn influences the field of photojournalism and its public perception and requires photojournalists to follow a "gradual process of digital-aesthetic refinement" (Solaroli 2015, p. 525) instead of too high a degree of optimisation or manipulation.

In the United States, the National Press Photographers Association (NPPA) has established a Code of Ethics that awarding institutions such as the Best of Photojournalism or Pictures of the Year International use as a guideline (Lough 2021, p. 306). The Code of Ethics (NPPA 2021) includes ten standards that "visual journalists and those who manage visual news production are accountable for upholding", including being accurate and comprehensive, providing context, and treating all subjects with respect (NPPA 2021). Concerning the discussion around editing photos, the Code of Ethics states that "editing should maintain the integrity of the photographic images' content and context. Do not manipulate images or alter sound in any way that can mislead viewers or misrepresent subjects" (NPPA 2021).

Seven further, more normative points should be followed by photojournalists ideally, such as defending "the rights of access for all journalists", or to "seek diversity of viewpoints" (NPPA 2021). The Code of Ethics concludes by stating that "visual journalists should continuously study their craft and the ethics that guide it" (NPPA 2021).

### 2.1.3. News Values in Photojournalism

Interestingly, the Code of Ethics does not explicitly mention a newsworthiness criteria for journalistic photos, although the selection of certain topics as newsworthy is a crucial part of the (photo)journalist's everyday work. "News values play a role, like how an understanding of the newsworthiness of Tiger Woods' comeback provided necessary contextualization of why a golf photo told an important story" (Lough 2021, p. 317).

Specific values that make information newsworthy—such as negativity, prominence, surprise—have been investigated by journalism scholars since the 1960s (see Östgaard 1965; Galtung and Ruge 1965). Among the twelve news values established by Galtung and Ruge (1965), the negativity factor received additional attention in journalism research as being a highly effective news value. First, there is often a broad consensus and thus unambiguity around the interpretation certain information as negative. Furthermore,

negativity commonly plays together with many other news factors, such as frequency, consonance, and surprise.

The news values have been updated gradually (Fretwurst 2008), including specific photo news values (Rössler et al. 2011; Godulla and Wolf 2016). As found by Rössler et al. (2011), photojournalists consider emotions, prominence, technique and surprise to be the most important photo news values, while to the recipients, the photo news values of aggression, technique, damage, and emotions are most important (p. 217). Additionally, the authors argue that photo news values are incorporated into a triangle of factors influencing the photo selection, with the personal dispositions of the photojournalists as well as their background knowledge also shaping the selective process (Rössler et al. 2011, p. 216). All three aspects are furthermore embedded into the editorial circumstances, which again influence the selection (Rössler et al. 2011, p. 216). Hence, in the broad field of photojournalism, not only the question of how (professional and ethical standards) photos are taken, but also which photos are taken and selected is of utmost importance.

### 2.1.4. Stereotypes in Photojournalism

The result of this process of carefully taking and selecting photos has an influence on how a topic is received by the audience. Referring back to the Code of Ethics (NPPA 2021), one of the ten standards is to "avoid stereotyping individuals and groups". In contemporary news, however, there is the danger that the "main use of photography is to represent news in a particular form, as a product that has little need for photographs beyond stereotypical illustrations" (Taylor 2000, p. 129). Research shows that in photojournalistic practice, stereotypes related to topics such as gender (i.e., McEntee 2018) or developing nations (i.e., Greenwood and Smith 2007) are reinforced rather than counteracted. This underlines the power of the professional actors orientating themselves within the realms of photojournalism, who frequently select photos for a bigger audience.

### 2.2. Professional Actors

Professional actors in the field of photojournalism are individuals and organisations who influence the field and orient themselves within the realms of this institution. First of all, this includes professional photojournalists themselves. "The field of photojournalism is more than just the images sent out to the world. It's made up of thousands of photographers who regularly define, shift, and redefine what it means to do photojournalism" (Lough 2021, p. 319).

These thousands of photographers "are increasingly in competition from both within (digital or multimedia journalists) and outside (citizens) the industry" (Ferrucci et al. 2020, p. 379). Since the field of photojournalism has been affected by "technology and business model changes", this has "profound impacts for photojournalists' development, roles, and responsibilities" (Thomson 2018, p. 803). Consequently, photojournalists find themselves employed in various ways, for example, as newsroom staff, freelancers, or photo editors (Thomson 2018, p. 803).

Alongside the originators of photos, there are also distributors and providers of journalistic photos, mainly press photo agencies such as Associated Press (AP), Agence France-Presse (afp) or Deutscher Depeschen Dienst (ddp). These agencies complement the work of the corresponding press (text) agencies and provide material highly topical to the events of the day (Wilke 2008, p. 63). Further actors include universal agencies and archives such as *Getty Images*, which offer more timeless material for journalistic coverage. These professional actors, engaging in the previously described field of professional photojournalism, are eligible to submit their selected photos to the World Press Photo Contest.

### 2.3. World Press Photo Contest

The World Press Photo Contest was first held in 1955, when the Photographers Association of the Netherlands celebrated their 25th anniversary by announcing a worldwide

photo competition; it was regarded as such a success that, five years and several contests later, the World Press Photo Foundation was founded. As "an independent, non-profit organization that has its home in Amsterdam, the Netherlands" (World Press Photo 2021e), it represents the leading international competition for journalistic photography (Godulla 2009; Hadland et al. 2016b). It does not mirror photojournalism in total, but specifically *professional* photojournalism (Godulla 2014, p. 63). As a non-profit organisation, its goal is not to achieve financial profit, but to create a forum to support the profession of photojournalism best in its development (Godulla 2015, p. 3). "It recognizes the best visual journalism of the last year, rewarding images and stories in eight categories"[1] (World Press Photo 2021f). Thereby, "(...) news photo contests act as consecrating institutions within the photojournalistic field, and they can play a normative role" (Solaroli 2015, p. 521). For example, by generally refusing any form of censorship and insisting on always showing the full and complete exhibition, the foundation is a prominent representative of the concept of press freedom (Godulla 2015, p. 3). As well as hosting exhibitions and competitions, the World Press Photo Foundation also acts as honorary organiser of further educational training (Godulla 2015, p. 4).

### 2.3.1. Judging Procedures at the World Press Photo Contest

In 2021, 74,470 images taken by 4315 professional photographers from 130 countries have been submitted to the contest. These numbers are typical for recent years. In order to select the award-winning photos, seven juries each consisting of three members shortlist the entries and have to undertake a radical reduction of material. These juries are devoted to the topics of environment, nature, contemporary issues, long-term projects news, portraits, and sports (Godulla 2015, p. 4), and the jury members are selected from the pool of aforementioned professional actors. Currently, the judging process lasts several weeks and has six voting rounds.

"The criteria for judging entries is a combination of news values, journalistic standards, and the photographer's creativity and visual skills" (World Press Photo 2021f). The judging process includes entry checks, manipulation reviews, and fact-checking (World Press Photo 2021f). Once the shortlisting is complete, a general jury consisting of six members and a chair, supported by a secretary, selects the winners. The understanding of photography as a visual language, its vocabulary and grammar, is modified and further developed each year through the changing jury (Godulla 2015, p. 5). The jury chair of 2021, Nayan Tara Gurung Kakshapati, emphasised the contests' purpose of joining "vital conversations on the politics of creating visibility for human and non-human conditions and defining moments that the world needs to pay attention to, deliberate on and remember" (World Press Photo 2021g).

### 2.3.2. World Press Photo of the Year

Each year, one photograph that "represents an event or issue of great journalistic importance that year" and displays "visual creativity and skills" is awarded the World Press Photo of the Year Award. Alongside the World Press Photo of the Year, there is also the World Press Photo Story of the Year award.

### *2.4. Dissemination*

The World Press Photo Contest would not be regarded as such an influential institution within professional photojournalism if it did not disseminate each year's results and award-winners and curate the data gathered from many years of the contest.

First of all, the winners are announced during an awards ceremony. Then, the photos are stored in a digital archive on the Foundation's website. Each year, the Foundation also publishes a yearbook, "featuring the most striking images and compelling stories" (World Press Photo 2021f) of the past year. Since it was first published in 1962, the book has become an established multiplier for press photography (Godulla 2015, p. 5). The winners are also included in the annual World Press Photo Exhibition, which travels across the globe from Italy to Rwanda to Australia. Joop Swart, as former president of World

Press Photo, emphasised these aspects in particular as "not only an important book, not only an imposing exhibition, but first and foremost an extremely valuable documentation of the time in which we live" (Swart 1988, p. 10). Communicating the results in such a manifold way not only shapes the professional field of photojournalism by defining standards and highlighting what professional photojournalism looks like, but also means they are observed by the public through various channels and means.

### 2.5. Public Communication

These channels and means of public communication are provided and shaped by media systems, which comprise the mass media of different countries "embedded in their social and political environment, which is also culturally—and nationally—shaped (Thomaß and Kleinsteuber 2011, p. 25). Thereby, media systems can differ in terms of press circulation, political parallelism, the development of journalistic professionalism, and the degree of state intervention (Hallin and Mancini 2004, p. 21). Since the World Press Photo Contest has a global reach, it is relevant to recognise the wide spectrum of different media systems that photojournalists submitting their photos to the contest are embedded and socialised in. Some submissions are made by photojournalists from countries with a lack of press freedom and strict state intervention, while others submit from more liberalised and democratic systems. Most countries, however, have both legacy media, such as print and broadcast media, as well as at least a certain occurrence of a digital mass media system at their disposal. The latter comprises digital publishing as well as social platforms, such as Facebook, Instagram, Twitter, or Snapchat. Through these channels—legacy media and newer digital publishing formats and platforms—information about the World Press Photo Contest can generally be distributed.

Legacy media, such as print and broadcast, cover topics traditionally, researching and editing information according to the rules of the specific format and then sending it out to the public; hence, "journalism is the key profession within the legacy media system" (Perusko et al. 2017, p. 2). They pick up the results of the World Press Photo Contest and communicate them through TV, radio and print to their consumers. Public communication in the digital sphere operates by other rules. Here, communication is more participative and discursive, with consumers of content also becoming producers of content, resulting in the construct of "prosumers", a term established by Toffler as early as 1980.

The aforementioned professional actors work within the realms of this range and take part in it through their news photography. Through public communication, they can initiate, shape and lead the discourse with the public about the World Press Photo Contest and its impact on professional photojournalism.

### 3. Literature Review: Analysing World Press Photos

Existing scholarly literature about World Press Photo in scientific databases such as SAGE Journals, Taylor and Francis Online, Emerald Insight, and Google Scholar is rather scarce, with only around two dozen articles that deal specifically with World Press Photo or are otherwise related to the topic. These will be outlined in the following; primarily, these articles are dealing with conditions of the World Press Photo Foundation and three different perspectives of the field of photojournalism: the topics displayed on the photos (1), the negativity displayed on the photos (2), as well as the underlying power structures of the competition (3). The conditions as well as the three identified perspectives are going to be introduced in the following and build the basic structure the study at hand. Finally, there are a few monographs relating to World Press Photo as well.

### 3.1. Conditions of the World Press Photo Foundation

To start with, the World Press Photo Foundation itself has conducted research into practices and standards of photojournalism and the working conditions of photojournalists (Campbell 2014; Hadland et al. 2016a, 2016b). Campbell's (2014) study examines the *integrity* of photojournalism by determining what practices and standards are undertaken

in relation to the handling of image manipulation in professional photojournalism and documentary photography today. Campbell states that manipulation occurs when significant changes are made to an image by adding or removing content, with the viewer of the image being deliberately deceived or misled. Minor adjustments such as limited cropping, toning, color adjustment, or conversion to grayscale are generally considered acceptable. The boundaries of where minor alterations end and manipulation begins, however, are blurring, and according to Campbell, a general consensus is lacking.

In addition to this, Hadland et al. (2016a) examine photojournalists' attitudes toward their work practices and the demands on their (technical) skills on behalf of the foundation. They surveyed participants in the 2016 World Press Photo Contest with the aim of obtaining knowledge about the conditions and skills of photojournalism practitioners. The sample amounted to 1991 respondents. Factors such as physical risks at work, self-sufficiency, the existence of other income, and also the skill requirements of photojournalists (such as the ability to produce videos) were examined. Results show a heavily male-dominated profession, with high levels of perceived risk, a rise in part-time work and an increasing requirement to also shoot video despite the photographers preferring to work with photo only. Generally, the survey respondents evaluated their job as an enjoyable activity.

Another study, conducted by Hadland et al. (2016b), was not carried out on behalf of the World Press Photo Foundation but in collaboration with it: *Photographers* who submitted their work to the contest in 2015 were invited to participate in a survey ($N = 1556$) which specifically focused on the risks to which the respondents felt they were exposed in their working environment. The main risks mentioned were physical (injury or death), an income that is unpredictable, a lack of provision for families, and the declining demand for photographic work. The assessment of these risks, especially the physical ones, was most strongly influenced by the respondents' background.

Mortensen et al. (2019) also draw on the World Press Photo survey but focus on the photographic subjects. Based on 15 interviews with people who were featured in photo essays that won international awards between 2013 and 2017 (National Press Photographers Association (NPPA) Monthly Clip Contest, the NPPA Best of Photojournalism Contest, and the World Press Photo Contest), the researchers examine factors influencing people's willingness to let journalists tell their story through a photo essay. According to the study, the protagonists' motivations were primarily to tell a different side of the story, inspire others, bring about change, or preserve moments through professional photography.

### 3.2. Perspective 1: Genres and Topics

Unlike the aforementioned contributions, Kedra (2016) examines the genres or classifications made within photojournalism. Based on a literature review and visual material from the analysis of a Polish daily newspaper, she creates a typology that identifies four photo genres: news photography, reportage photography, portrait photography, and illustrative photography. The author concludes that genres often tend to merge and thus any attempt at classification always leads to generalisations. Kędra mentions World Press Photo or the Contest in her argumentation only insofar as she refers to the classification made in the yearbook and thus the genres made within the contest for photojournalism in her description of the research subject.

Covering the practice of *sporting activities* as a specific strand, Haynes, Hadland and Lambert's study (Haynes et al. 2017) examines the status quo of the profession of sports photography based on a survey conducted in collaboration with the World Press Photo Foundation. The sample consisted of 713 respondents. The researchers determined that challenges specific to sports journalism include limited access to the scene of the action (e.g., events, sports venues), commercial licensing from media partners, and increasing management of image rights. Most photographers cover a wide range of sports, especially those who derive much of their income from sports. As with Hadland and Barnett (2018), a declining female representation can also be noted within sports photography.

Considering the sports photography category of the World Press Photo contest, Marfil-Carmona et al. (2018) examine the winning photos of 2017, with the aim of determining what understanding of sports is conveyed to the public through press photography. The researchers find that primarily aesthetics and presentation characteristics are conveyed, with the emphasis of the photography on a professional and competitive dimension. Marfil-Carmona et al. (2018) conclude that, based on the photographs, sports are only depicted in connection with the professional elite, and are thus not associated with everyday activities or a healthy lifestyle.

In his study, Arifin (2019) examines the representation of public space or nature based on the *photos from Jakarta* submitted by Peter Bialobrzeski, which were awarded second place in the nature category of the 2009 World Press Photo Contest. By means of image analysis, the researcher concludes that public spaces are "green", but at the same time he limits that interpretation, for a better picture of the present surroundings should not only refer to the photos, but should take place in the field.

Hasan's (2015) study explores the photography of natural catastrophes by content-analysing images of the eruption of Mount Merapi. These images were taken by Kemal Jufri in 2011 and were awarded second place in the stories category of the World Press Photo contest. Hasan also applies the technique of semiotic image content analysis. Here, the photos (*N* = 4) primarily show the suffering of the people who were affected by the eruption, as well as the collective help of Indonesians for the victims of the catastrophe.

König (2010) focuses on Africa in her elaboration and examines the visual representation of the continent in the press image. Her sample consists of World Press Photos of all categories and price levels that focus on Africa and its inhabitants and were published between 1999 and 2009. König first examines the photos (*N* = 102) by means of a quantitative content analysis, including general categories such as the photographer's origin or placement, as well as formal (form of presentation, perspective, setting size, colour and effects) and content-related characteristics (topic, person- or subject-centredness, number of people, gender, actors, clothing, sensitivities, emotions and violence in the image). The author concludes that visual stereotypes are increasingly used: Nature photographs primarily show positive aspects and romanticise, while photographs that focus on the population tend to convey an image of political, economic, and social instability.

### 3.3. Perspective 2: Negativity

Hofmann's (2015) dissertation deals with the topic of photographic eye-witnessing and analyses ethical and aesthetic aspects of World Press Photo in this regard. Through access to the World Press Photo Contest archive, the researcher examines selected representations of war and violence as well as flight and migration in the period from 1989 to 2005. The sample is composed of the yearbooks and newsletters published by World Press Photo. Hofmann analyses the photos in the ethical-aesthetic dimensions of authority and anonymity, closeness and distance, and instantaneity and duration, in order to show, the status of photographic eye-witnessing and its development since the beginning of the competition. The author comes to the conclusion that photographic eye-witnessing is no longer a medium of truth. Rather, it is to be understood as a visual form of communication that articulates and demands responsibility for a finding.

Below's (2010) master's thesis explores photojournalism within war reporting, specifically examining the characteristics of war- and crisis-related photographs and the depiction of victims in conflicts. The author undertakes a qualitative content analysis in which she examines six World Press Photos of the Year that depict the object of study (war). To capture changes in the photographic representation of conflict, Below analyses one photo from each decade since the contest began. The author concludes that victims tend to be presented from a distance and that perpetrators are particularly often depicted in connection with the military. Features such as colourlessness, drabness, dust, smoke, debris, and fire contribute to frame the image accordingly. Negative emotions and depictions are constant throughout

the decades studied and, according to Below, are more likely to capture people's attention than positive emotions.

Smith (2015) argues in his master's thesis that the origins of the Arab Spring have been sufficiently studied but press photography in this regard has not. Therefore, the goal of his research is to show what patterns and characteristics are common to photographs of the *Syria conflict* that have received an award in the field of photojournalism (National Press Photographers Association, World Press Photo Contest, and Pulitzer Prize). Smith undertakes a qualitative investigation by conducting a semiotic image content analysis on the selected photographs (*N* = 99). The author identifies a frequent occurrence of the roles of "victim" and "outsider" among the civilians depicted, as well as a portrayal of Syrian rebels as "loveable losers" of the conflict. These roles, according to Smith, distinguish the photographs from earlier depictions of the two Gulf wars, with both the civilian population and the rebels perceived more empathetically.

Human suffering is an equally common theme of World Press Photos but can have different orientations. For example, Mannevuo (2014) examines the portrayal of *malnutrition*, while Ravel's (2013) research focuses on the depiction of *religious suffering*. Mannevuo (2014) engages with the photography of malnourished children in her study. She conducts a qualitative descriptive analysis of five World Press Photos of the Year (1974, 1980, 1992, 2001, and 2005) depicting this subject. The aim of the study is to show what features distinguish the images and how these features might affect the viewer. Mannevuo concludes that the images trigger questions about equality, violence, and also personal concern in the recipient.

In her multi-method study, Ravel (2013) analyses World Press Photos of the Year taken in the course of different political conflicts (Kosovo, Algeria, and Yemen), but in contrast to the category of war and conflict, these do not illustrate the violence taking place, but rather the *(religious) suffering* resulting from it. The photographs in the sample date from 1991, 1997, and 2011 and depict Muslim women in intimate moments of intense suffering.

In summary, the diverse studies about photojournalism under the umbrella of World Press Photo show a range of different facets with which this field is interwoven. In addition to the working conditions and skills of the photographers, the presentation of topics, negativity and suffering are also worth analysing and give insights into a changing field as well as the portrayal of countries and continents around the world.

### 3.4. Perspective 3: Power Structures

As well as examining the photographers' working conditions, practices and displayed topics, several researchers have placed their focus on the analysis of the photographs submitted to or awarded in the competition in terms of different social, political and continental contexts. Therefore, in these studies, the photojournalists and their origins as well as the countries or continents depicted are the focus of the investigation and offer insights from different perspectives. The specific role of photo contests has been investigated by Zarzycka and Kleppe (2013), who examine the repeated occurrence of typical figurative expressions in photographs that won awards in the World Press Photo Contest from 2009 to 2011. These expressions include, for example, grieving people, civilian soldiers, or witnesses to atrocities. The authors argue that photography contests such as the World Press Photo Contest can influence the process of generic understanding of war and catastrophe based on their images.

An essential question arising in the presentation of war and catastrophe is how a country or continent is portrayed and by whom. König (2010) argues that the press photographs studied in her elaboration show an image of Africa that is reduced to a few negative themes and is thus one-sided. According to König, the dominance of European photographers suggests a Eurocentric view. Positive political, economic and social developments are hardly picked up, and only rarely is participation in a regulated social life conveyed in a picture. König views such portrayals as necessary for more balanced picture reporting.

Likewise, Li (2019) deals with a specific group of practitioners, focusing exclusively on Chinese photographers. In particular, she examines the influence of Western values and interactions on their work. The study focuses on the development of professional photography in China from 1937 to 1988, using archival research and interviews. The author only introduces a reference to the World Press Photo Contest in her elaboration insofar as she describes the contest as a major influencing factor and reference point for the "Westernization" of Chinese photography. Li argues that prior to the influence of Western values on Chinese photography, numerous images were submitted but never awarded prizes. Only with increasing interaction with the West (especially from 1978–1988) could a growing influence of Western norms and values be noted, and the previously isolated Chinese photography changed to a more open and diverse culture.

In his dissertation, Godulla (2009) examines all photographs ever to be exhibited by World Press Photo in the period 1955 to 2006 ($N$ = 9647) using quantitative content analysis. In addition, a qualitative investigation of the 49 overall winning photos published to date is conducted. The study also represents the first full survey and evaluation of all photos on an empirical basis. While the qualitative analysis refers concretely to the picture object, the quantitative analysis aims to provide the genesis of an abstract overview. In particular, the continental affiliation of groups of actors such as jury members, participants, and competition winners is coded, and the content-related criteria of the photographs (e.g., themes depicted, negativism, technique, and narration) are recorded. In contrast to other studies, which also refer to social contexts and phenomena, this study is the only one to systematically analyse the development status of the respective countries at the time the photos were taken, with reference to the Human Development Index (HDI). Godulla (2009) found that a large proportion of the 49 overall winning photographs examined were from crisis areas, while the photographers were almost always from Western countries. In addition to a photographer's continental affiliation, the economic and social development of his or her country of origin also plays a decisive role in the chances of winning.

Liu (2013), also dealing with the professionalism of photographers in her study, takes a more indirect approach through examining the winning photos ($N$ = 148) in the environment category from 1992 to 2011. The researcher analyses their journalistic objectivity and the media's claims to professionalism of both industrialised and developing countries, as well as non-governmental organisations (NGOs). In doing so, the findings are related to ideologies and power dynamics in both journalism and global political developments. Strikingly, developed countries dominated the discourse, but journalistic objectivity remained intact. The findings also show that the demands for objectivity of the image did not hinder the depth of detail and complexity of the photographs, as this could be achieved through storytelling and the use of multiple shots, among other methods.

Focusing specifically on the role of female photographers within the profession, Hadland and Barnett (2018) surveyed participants of the 2015 and 2016 World Press Photo Contests ($N$ = 3500), of which 545 were female. The results were then compared and related to their male counterparts. By means of a descriptive statistical analysis, it was determined that female photographers more often than male photographers face more demanding circumstances while practicing their profession. Their professional qualifications are also related to a higher degree and better overall level of photography education. Nevertheless, female respondents more often described their financial situation as difficult and indicated lower income brackets than their male counterparts. The underrepresentation of women in photography is expected to continue or even increase, according to Hadland and Barnett (2018).

Gender roles are not only examined in relation to the profession (Hadland and Barnett 2018), but also explored by Thalwitzer and Throm (2013) in relation to the people depicted in the photographs. The authors examine the mass media (de)construction of *gender relations in Yemen* using the 2011 World Press Photo Contest as an example. Thalwitzer and Throm discuss the oppression of women in the Arab Spring based on a descriptive analysis of the winning photo. According to the qualitative research, the photo depoliticises the

Arab Spring because it shows a private scene. It enforces stereotypical gender roles and at the same time shows a Christian reference.

### 3.5. Summary

In summary, the publications could be differentiated in terms of contributions dedicated to the photos and their production conditions, topics, displayed negativity, and research about the power structures of photo contests. The thematic analyses tend to have a qualitative approach in their methodology. A limitation of almost all publications mentioned is the sampling—at least when findings on photojournalism as a profession are obtained. Since the sample almost always focuses on images and actors of the World Press Photo Contest, only photographers who participate in contests are examined. For the study of the World Press Photo Contest as a research object, however, this sampling is appropriate. Another way to expand the sampling is to look at other competitions in the field of photojournalism, but this is only done in two of the studies presented (Mortensen et al. 2019; Smith 2015). However, it should be noted here that other competitions sometimes take a different approach to selecting the winning photos and therefore there is no real comparability.

In addition, only a few studies to date have looked at the effect of World Press Photos on the audience (including Ravel 2013; Mortensen et al. 2019). This is an area future research could explore. It is also interesting to note that studies examining the profession are often mirrored in the focus of scientific articles with thematic analysis and vice versa. Accordingly, it can be concluded that both the profession and the photographs arising from this profession cannot be considered separately and might influence each other. It is important to note that in this review only German and English language contributions have been presented, and that there are certainly numerous studies in other languages due to the global nature of the contest.

Additionally, complementing the insights into World Press Photo found in academic journals, a handful of monographs have been published to mark relevant anniversaries of the institution as well, which should not got unmentioned. Evans published two books (Evans 1981, 1985), called *Eye Witness* and *Eye Witness* 2, in which he takes the reader through 25 and 30 years of World Press Photo respectively. Featuring twelve texts about professional photojournalism, for instance, war photojournalism, Meijer and Swart (1988) cover *The Photographic Memory* in their book of the same name. Including essays as well, Mayes (1995) published the book *The Critical Mirror* on the 40th anniversary of the contest, focusing on the morphology of the press photo itself. Ten years later, Panzer (2005) covers the journalistic photograph in its published context *in Things as they are*, raising awareness of photo manipulation and altered messages through headlines and picture texts.

### 3.6. Derivation of Reserarch Questions

Based on the portrayed findings about World Press Photo and the World Press Photo Contest, and thereby considering the embedding of the topic within the realms of professional photojournalism, three research questions arise:

First, connecting the dots of the first part of the literature review, which focused on the submitted photos, a detailed and comprehensive analysis of the topics of all World Press Photos of the Year is required. Assessing these topics will not only give insights into the development of professional photojournalism in the past 62 years, but also build the foundation for further analyses. Hence, research question 1 (RQ1) asks:

**RQ1.** *What topics are depicted in World Press Photos of the Year?*

Having assessed these topics, a specific focus on the news factor of negativity is of timely interest, considering the history of remarkable but mainly negative World Press Photos of the Year and the groundbreaking notion of positivity in the 2021 winning photo. It is not only the presence of negativity that is of interest, but also different expressions and

facets of negativity as well as its development over time and in different continents. Hence, research question 2 (RQ2) asks:

**RQ2.** *In what way do World Press Photos of the Year correspond to the news factor of negativity and how does this manifest itself?*

Linked to the introduced body of literature that investigates the social and political implications of the contest, especially when depicting countries in a certain way, and considering the previously introduced danger of stereotyping in photojournalism, an investigation of the countries and photojournalists submitting photos and the countries displayed in the images is expected to reveal patterns and insights into the broader cultural and economic context of professional practices. In order to do this comprehensively, the whole time span of 62 years and the countries' levels of development are considerable variables. Hence, research question 3 (RQ3) asks:

**RQ3.** *What is the relationship between the country depicted and its level of development on the one hand, and the photographer's country of origin and its level of development on the other?*

The research questions will be answered in the following through a qualitative analysis of all World Press Photos of the Year awarded between 1955 and 2020. At the same time, this will provide the basis for the subsequent derivation of the hypotheses. In the following, an overview of the further research process and the composition of the samples analysed.

### 4. Method: Qualitative and Quantitative Analyses of World Press Photos

Figure 2 illustrates the structure of the research project and reveals how each analysis provides the basis for the subsequent analysis. The literature review helped to identify the three main foci of the displayed topics (RQ1), the degree of negativity (RQ2), and certain power dynamics (RQ3). In order to be able to answer these research questions, first a qualitative content analysis of all World Press Photos of the Year available until 2020 ($N = 62$) is conducted. The results of this first analysis produce hypotheses, again related to the topics, negativity and power structures, which are tested with the help of two quantitative analyses. The first focuses on all photographs ever published in the available World Press Photo Yearbooks ($N = 11,789$), assessing the topics and their levels of negativity; the second on the power structures by assessing the origin of all jury members ($N = 686$) and participants ($N = 1,652,800$) and the Human Development Index (HDI) of these countries, as well as the placement of the photographs (2347).

For the first period of the analysis (first decade, 1955–1959), only the number of jury members is available. The number of participants and rankings can only be taken into account in the analysis beginning with the second decade (1960–1969), as these were not published prior to this period. It should be noted that all photos that received a ranking in the competition were already taken in the previous year. For example, all photos that received an award in 2020 were taken in 2019. Therefore, in the following, different periods of time occur, ranging from either 1955 or 1960 to either 2019 or 2020. The two perspectives of topics and negativity, which are depicted in the photos examined, can likewise only be evaluated beginning with the period from 1960 onwards, since no data are available prior to this period.

The paper's structure follows the structure of Figure 2; in the following section, the qualitative analysis is outlined, resulting in hypotheses which are then tested in the subsequent quantitative analyses.

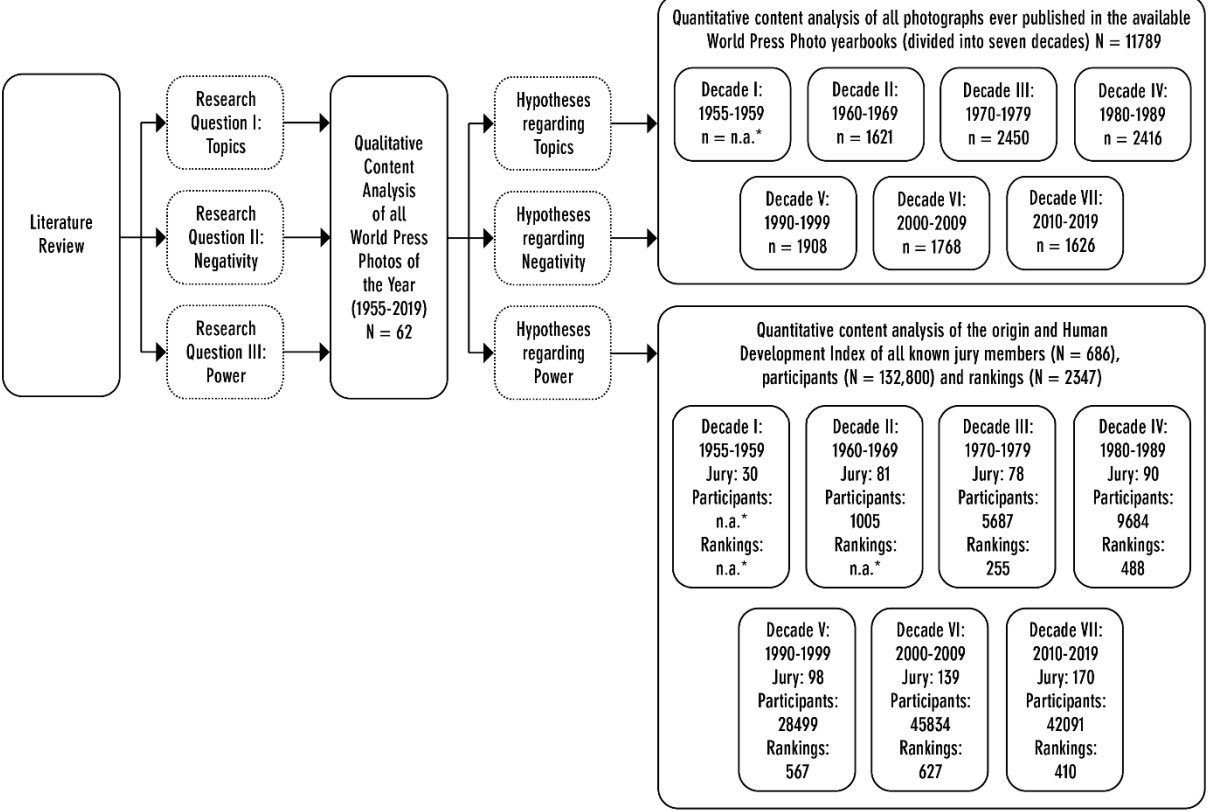

**Figure 2.** The Structure of the Research Project[2].

## 5. Qualitative Content Analyses of All World Press Photos of the Year (1955–2019)

In order to gather insights into the topics displayed (RQ1), the negativity displayed (RQ2), and the power dynamics underlying World Press Photos (RQ3), as well as the development of all these aspects over time, a qualitative content analysis of a selection of these photos is most appropriate. As a systematic, intersubjectively revisable method with which material quantities can be managed in a qualitative, interpretative manner, the qualitative content analysis makes it possible to capture latent meanings, too (Mayring and Fenzl 2014, p. 543). It offers the possibility of using advantages of the quantitative content analysis and applying them to a qualitative interpretation. The systematic approach results from the creation of categories that come together in a category system as the main instrument of analysis (Mayring and Fenzl 2014, p. 545). According to Mayring (2015), there are three different approaches to qualitative content analyses, namely summary, explication and structuring (Mayring 2015, p. 65ff). The summarising content analysis seems the most appropriate for analysing World Press Photos in light of the posed research questions, as it summarises the material gradually and reduces it to its main information. Afterwards, the generated inductive categories can be used for a subsequent quantitative analysis of these photos.

The "World Press Photo of the Year", as previously described, represents an expressive sample for the qualitative analysis at hand. Hence, all 62 World Press Photos of the Year that were awarded from 1955 until 2020 underwent a qualitative content analysis. Analysing the topics, negativity and power relations displayed in these photos generates an overview of categories as well as sub-categories for the further quantitative analysis of a bigger sample size. The specific methodological approach to each of the three aspects (topics, negativity and power relations) as well as the respective results are described in the following.

*5.1. Perspective 1: The Displayed Topics in World Press Photos of the Year (RQ1)*

In order to structure and analyse the displayed topics, an image sorting study was carried out. Image sorting studies aim to document the process, structure, and result of the category creation process (Geise and Lobinger 2016, p. 508). Thereby, information units are categorised based on their perceived characteristics and assigned to groups with similar levels of abstraction; in this way, they are ascribed a reconstructable meaning and their relation towards other objects can be revealed (Geise and Lobinger 2016). Image sorting studies can be differentiated between two different methods: the Q-Sort (Petersen and Schwender 2011, p. 322), in which a smaller sample size of elements is brought into a specific order and interpreted according to preferences, and the Card-Sorting (Petersen and Schwender 2011), in which elements are sorted and grouped and later interpreted. Since the latter can combine a bigger sample size, it is most applicable for the study at hand. Image sorting studies are especially fruitful when analysing visual communication, since the rather subjective interpretation of an image can be effectively implemented with this methodological approach (Geise and Lobinger 2016, p. 509). Two independent coders analyzed the material accordingly, and then compared their results and the inductively derived topical groups. In a subsequent discussion about the similarities and discrepancies of both their inductive coding processes, the two coders found six main topics in consensus.

When sorting all 62 World Press Photos of the Year since 1955, the results show six main topics these photos could be assigned to. The most frequent topic found in the photos is conflict (29), which thematises fighting with armed force between states or non-state collectives. As well as the direct depiction of armed combat, all forms of terrorism, such as bombings, and long-term consequences of war also belong under the umbrella term of conflict. For example, the Vietnam war and numerous civil wars have been the topic of these photos. The second most frequent topic is politics (14), referring to political processes such as official and procedural events, election campaigns, or demonstrations. A further eight photos can be grouped under the term catastrophe, where people affected by a natural or technical catastrophe, or the consequences of such happenings, have been captured, such as earthquakes in Armenia, Iran, or Turkey. A further seven photos deal with topics such as immigration and homophobia, thus focusing on social problems or stigmas, and can be summarised under the topic of social documentary. Two of the photos captured a *sports* theme, which could refer to amateur as well as professional sports, and two a *society* topic, thematising happenings that range from the everyday life of people to cultural aspects to tabloid topics. The named topics and their respective years of publication can be found in Table 1.

These inductively derived categories—conflict, politics, catastrophe, social documentary, society, and sports—can now be used as categories for a quantitative analysis of a greater number of photos. Based on the commonly encountered topics, these categories have been further expanded in the quantitative analysis through considering the categories found in the World Press Photo Yearbooks, which have been subject to change throughout the years. Going through all yearbooks from 1955 until 2020, the categories nature & environment, art & culture, and science & technology were added. Furthermore, the category religion was added, since it became apparent that the inductively derived category society could not in full map the associated topics, of which one apparent stream was the religious one. Furthermore, the prevalence of displayed conflict once again underlines the necessity to examine negativity displayed in World Press Photos, which will be done in the following.

**Table 1.** World Press Photos of the Year 1955–2020 sorted by topics.[3]

| Conflict (29) | Politics (14) | Social Documentary (7) |
|---|---|---|
| 1956: Coming home from World War II | 1957: Racial segregation | 1986: AIDS |
| 1962: El Porteñazo uprising in Venezuela | 1960: Assassination of Inejirō Asanuma | 2000: Immigration to the United States |
| 1964: Cyprus Conflict | 1963: Suppression of Buddhists in Vietnam | 2008: Subprime mortgage crisis |
| 1965: Vietnam War | 1969: The Troubles | 2010: Taliban treatment of woman |
| 1966: Vietnam War | 1973: Coup in Chile | 2013: African migrants |
| 1967: Vietnam War | 1977: Apartheid | 2014: Homophobia in Russia |
| 1968: Vietnam War | 1978: Sanrizuka struggle | 2018: Immigration policy of Donald Trump |
| 1972: Vietnam War | 1981: 23-f coup attempt in Madrid | |
| 1976: Lebanese Civil War | 1987: Election in South Korea | **Society (2)** |
| 1979: Fall of the Khmer Rouge in Cambodia | 1989: Tiananmen Square Massacre | 1971: Bank robbery in Saarbrücken |
| 1980: Famine in Karamoja, Uganda | 2009: Iranian presidential election | 1975: Fire escape collapse in Boston |
| 1982: Lebanon War | 2011: Protests in Yemen, Arab Spring | |
| 1990: Kosovo conflict | 2017: Crisis in Venezuela | **Sports (2)** |
| 1991: Gulf War | 2019: Sudanese coup d'état | 1955: Motorcycle racing in Denmark |
| 1992: Somali Civil War | | 1958: Football in Czechoslovakia |
| 1993: Palestinian territories | **Catastrophe (8)** | |
| 1994: Rwandan genocide | 1974: Sahel famine, Niger | |
| 1995: First Chechen War | 1983: Earthquake in Turkey | |
| 1996: Angolan Civil War | 1984: Bhopal disaster | |
| 1997: Algerian Civil War | 1985: Armero volcano disaster | |
| 1998: Kosovo conflict | 1988: Earthquake in Armenia | |
| 1999: Kosovo War | 2002: Earthquake in Iran | |
| 2001: Refugees in Afghanistan | 2004: Indian Ocean earthquake | |
| 2003: Iraq War | 2005: Niger food crisis | |
| 2006: Lebanon War | | |
| 2007: Afghanistan War | | |
| 2012: Operation Pillar of Defense | | |
| 2015: European migrant crisis | | |
| 2016: Assassination of Andrei Karlov | | |

*5.2. Perspective 2: Types of Negativity in World Press Photos of the Year (RQ2)*

In order to analyse the negativity displayed in the 62 World Press Photos of the Year, a two-step approach was followed. First, the photos were sorted by types of negativity, this time specifically focusing on their displayed negative context, such as violence or accidents. Second, and in order to operationalise the negativity aspect, negativity was inductively divided into four concrete measurable categories in the course of the analysis: peril, emotional suffering, physical suffering, and death. Peril was present when the photo displayed a person in danger, emotional suffering when the photo displayed a person in emotional pain, and physical suffering was coded when the photo displayed a person in bodily pain. Death was coded if the photo showed a corpse or if there were clear references to dying, for example in the form of coffins or graves. For all four types of negativity, only the picture itself was decisive, in order to ensure that there was no room for interpretation in the identification process. Every photo was analysed for the presence of these four elements; whenever one element was present, it was coded with a plus sign (+). Whenever it was not present, it was coded with a minus sign (−). For example, a photo with the coding (+/+/+/−) shows peril, emotional and physical suffering, but not death. Only five photos could not be assigned to either one of these categories, hence, they did not contain any of the four aspects of negativity. The detailed distributions of the four characteristics of negativity for every photo can be found in Table 2.

**Table 2.** World Press Photos of the Year 1955–2020 (Negativity) (see Note 3).

| Violence (40) | Misery (8) | Suffering (1) |
|---|---|---|
| Peril (20/40), Emotional Suffering (24/40), Physical Suffering (11/40), Death (7/40) | Peril (2/8), Emotional Suffering (5/8), Physical Suffering (5/8), Death (2/8) | Peril (0/1), Emotional Suffering (1/1), Physical Suffering (1/1), Death (0/1) |
| 1956 (−/+/−/−), 1957 (−/+/−/−), 1960 (+/+/−/−), 1962 (+/+/+/−), 1963 (+/−/+/−), 1964 (−/+/−/−), 1965 (+/+/−/−), 1966 (−/−/−/+), 1967 (+/−/−/−), 1968 (+/+/−/−), 1969 (+/−/−/−), 1971 (+/−/−/+), 1972 (+/+/+/−), 1973 (+/−/−/−), 1976 (+/+/−/−), 1977 (+/+/−/−), 1978 (+/−/+/−), 1981 (+/−/−/−), 1982 (−/−/−/+), 1987 (−/+/−/−), 1989 (+/−/−/−), 1990 (−/+/−/+), 1991 (+/+/+/+), 1993 (−/−/−/−), 1994 (−/+/+/−), 1995 (−/+/−/−), 1996 (−/−/+/−), 1997 (−/+/−/−), 1998 (−/+/−/−), 1999 (−/+/+/−), 2003 (−/+/−/−), 2006 (−/+/−/−), 2007 (+/+/+/−), 2008 (+/−/−/−), 2010 (−/−/+/−), 2011 (−/+/+/−), 2012 (−/+/−/+), 2016 (+/−/−/+), 2017 (+/−/−/−), 2018 (−/+/−/−) | 1974 (−/+/+/−), 1979 (−/+/−/−), 1980 (+/−/+/−), 1992 (−/+/+/+), 2000 (−/−/+/−), 2001 (−/−/−/+), 2005 (−/+/+/−), 2015 (+/+/−/−)<br><br>**Accident (8)**<br>Peril (3/8), Emotional Suffering (6/8), Physical Suffering (1/8), Death (5/8)<br>1955 (+/−/−/−), 1975 (+/+/−/−), 1983 (−/+/−/+), 1984 (−/−/−/+), 1985 (+/+/+/−), 1988 (−/+/−/+), 2002 (−/+/−/+), 2004 (−/+/−/+) | 1986 (−/+/+/−)<br><br>**None (5)**<br>1958, 2009, 2013, 2014, 2019<br><br>**Total (62)**<br>Peril (25/62), Emotional Suffering (36/62), Physical Suffering (18/62), Death (14/62) |

Results of the topic sorting show four specific aspects of negativity, the most frequently featured being violence (40), followed by misery (8), accident (8), and suffering (1). Photos were assigned to violence when they displayed some kind of (applied) force, to misery when they displayed a hardship such as hunger or poverty, to accident when they displayed a specific, fateful event, and to suffering when they displayed a negative topic related to the mere grounds of being human. This includes, in particular, the fact that people grow old and can suffer from diseases due to their biological nature.

Coding the four above-mentioned characteristics of negativity (peril, emotional suffering, physical suffering, death) shows a dominance of emotional suffering, which occurs in more than half of the photos (36). Peril was present in 25 of the 62 photos, while physical suffering and death were displayed in 18 and 14 photos respectively.

These inductively derived categories can now serve as the basis for the subsequent quantitative analysis, which can examine for the presence of violence, misery, accident, and suffering, as well as for the specific characteristics of negativity (peril, emotional and physical suffering, death).

The individual specifications of the categories of negativity were inductively derived from the material and are shown in Table 3. A distinction is made between potential or acute as well as abstract, concrete or massive presentations of peril, emotional suffering, physical suffering and death and is explained by definitions.

**Table 3.** Definition of characteristics of negativity.

| Peril | |
|---|---|
| Potential | The photo depicts at least one person who is in a potentially life-threatening situation. The death of one or more persons in the near future is imaginable, but not mandatory (e.g., people in a war or catastrophic area, in the midst of violent outrages, or with serious injuries or illnesses). |
| Acute | The photo depicts at least one person who is in a highly life-threatening situation. Death of one or more people in the near future is likely and will probably occur soon after the photograph is taken (e.g., people facing a shooting commando, fleeing from a lynch mob, or with fatal injuries or illnesses). |
| **Death** | |
| Abstract | There are no dead people depicted in the photo. Instead, it indirectly thematizes death by showing, for example, a corpse covered by a towel, a closed coffin or an open grave. |
| Concrete | The photo shows dead people with no or only minor injuries. On first glance, one could get the impression that they are sleeping. |
| Massive | The photo shows dead people who may have serious injuries (e.g., bloody gunshot wounds, mutilations, signs of decomposition) or show facial expressions resembling fear or horror. Abuse of a corpse also justifies the use of this category. |
| **Emotional suffering** | |
| Concrete | There is at least one person in the photo who subliminally expresses feelings such as fear or sadness. |
| Massive | There is at least one person in the photo expressively showing feelings such as fear or sadness. |
| **Physical suffering** | |
| Concrete | There is at least one person in the photo who is suffering from noticeable injuries that are not life-threatening and not permanent. |
| Massive | There is at least one person in the photo who is suffering from serious injuries that are potentially life-threatening and permanent. |

*5.3. Perspective 3: The Power Dynamics Underlying World Press Photos of the Year (RQ3)*

In order to analyse the power dynamics underlying World Press Photos of the Year, each of the 62 photos was examined for the country shown as well as for the country of origin of the photographer. Acting from the assumption that some countries might be shown more frequently than others, as well as photojournalists from some countries or areas of the world being more often awarded with the World Press Photo of the Year than others, this analysis is expected to reveal insights into underlying power dynamics.

In order to further examine this aspect, the Human Development Index (HDI) was included for all photos that received awards since 1990. The Human Development Index is "a summary measure of average achievement in key dimensions of human development: a long and healthy life, being knowledgeable and having a decent standard of living" (United Nations Development Programme, UNDP 2020). Next to measures of a country's economy, such as the average income, the HDI also considers life expectancy and the average duration of education in years. Hence, it includes aspects of education and health and thus allows for a more nuanced assessment when it comes to investigating the overall situation of a country with regard to power dynamics. The HDI was only integrated for all photos since 1990 as the formula for calculating the HDI has changed over the years; since 1990, it has been calculated using the described variables.

Results show the most frequently displayed continent is Asia with a presence in 29 of 62 photos. Among these images, ten were shot by Asian photographers, while nine each were shot by North American and European photographers. Only one was shot by an African photographer. The situations depicted in the photos happened in Europe in 13 of the 62 cases, with ten shot by European photographers and three by North American photographers. Africa was depicted in ten photos, shot by three African, five North American and two European photographers. Finally, in six cases the country shown was

North America, in five cases shot by North American photographers and once by an Asian photographer. The countries depicted for each continent are shown in Table 4, where the first country mentioned refers to the country shown, and the second refers to the country of origin of the photographer. As the table shows, there is a prevalence of Asian countries shown in the photos, and a prevalence of the country of origin of the photographer being one with a high or very HDI.

**Table 4.** World Press Photos of the Year 1955–2020 (Country shown/Country of origin of the photographer) (see Note 3).

| Asia (30) | Europe (13) | North America (6) |
|---|---|---|
| Asia (10): | Europe (10): | North America (5): |
| 1960 (Japan/Japan) | 1955 (Denmark/Denmark) | 1957 (USA/USA) |
| 1965 (Vietnam/Japan) | 1956 (West Germany/West Germany) | 1975 (USA/USA) |
| 1966 (Vietnam/Japan) | 1958 (Czechoslovakia/Czechoslovakia) | 2000 (USA ****/USA ****) |
| 1972 (Vietnam/Vietnam) | 1964 (Cyprus/Great Britain) | 2008 (USA ****/USA****) |
| 1978 (Japan/Japan) | 1969 (Great Britain/West Germany) | 2018 (USA ****/USA ****) |
| 1983 (Turkey/Turkey) | 1971 (West Germany/West Germany) | Asia (1): |
| 1984 (India/India) | 1981 (Spain/Spain) | 1986 (USA/Israel) |
| 2002 (Iran **/Armenia **) | 1990 (Yugoslavia ***/France ****) | |
| 2004 (India */India *) | 1999 (Albania **/Denmark ****) | **South America (4)** |
| 2016 (Turkey ****/Turkey ****) | 2015 (Hungary ****/Denmark ****) | South America (3): |
| North America (9): | North America (3): | 1962 (Venezuela/Venezuela) |
| 1963 (Vietnam/USA) | 1988 (USSR/USA) | 1973 (Chile/Chile) |
| 1968 (Vietnam/USA) | 1995 (Russia ***/USA ****) | 2017 (Venezuela ***/Venezuela ***) |
| 1979 (Thailand/USA) | 1998 (Serbia ***/USA ****) | Europe (1): |
| 1982 (Lebanon/USA) | | 1985 (Colombia/France) |
| 1987 (South Korea/USA) | | |
| 1989 (China/USA) | **Africa (10)** | |
| 1991 (Iraq */USA ****) | Africa (3): | |
| 1993(Palestinian Territories/Canada ****) | 1977 (South Africa/South Africa) | |
| 2006 (Lebanon/USA ****) | 1997 (Algeria **/Algeria **) | |
| Europe (9): | 2019 (Sudan */Japan ****) | |
| 1967 (Vietnam/Netherlands) | North America (5): | |
| 1976 (Lebanon/France) | 1974 (Niger/USA) | |
| 2001 (Pakistan */Denmark ****) | 1992 (Somalia/USA ****) | |
| 2003 (Iraq **/France ****) | 1994 (Rwanda */USA ****) | |
| 2007 (Afghanistan */Great Britain ****) | 2005 (Niger */Canada ****) | |
| 2009 (Iran ***/Italy ****) | 2013 (Djibouti */USA ****) | |
| 2011 (Yemen */Spain****) | Europe (2): | |
| 2012 (Palestinian Territories ***/Sweden ****) | 1980 (Uganda/Great Britain) | |
| 2014 (Russia/Denmark ****) | 1996 (Angola/Italy ****) | |
| Africa (1): | | |
| 2010 (Afghanistan */South Africa **) | | |

Note: HDI at the time the photo was taken since 1990 as far as applicable. * low, ** medium, *** high, **** very high (UNDP 2020).

Summarising the results of the qualitative analysis of all World Press Photos of the Year (*N* = 62), one main observable pattern is that photographers from North America (including Canada) and Europe lead the field, with photographers from each continent having 22 winning photos. Therefore, photographs from highly or very highly developed countries dominate the competition when considering the HDI. The discrepancy is most obvious in the case of Africa, which features less in the role of submitting photos, and more in the role of providing scenes and events for photographs that are submitted. All in all, the winning photos show the world through the lens of the developed western world, a result which, referring back to the previously introduced practice of stereotyping, should be observed critically.

Another pattern becomes obvious in the news value of negativity. In only five of the 62 photos is no component of negativity shown, with four of these five being awarded since 2009, which might indicate a recent turn towards more positive or balanced photos, a trend that the winning photo of 2021 also joins. Taking emotional suffering and physical suffering together, these components of negativity are displayed in 52 photos, while the sample of 62 images also contains 14 photos of dead bodies or clear thematisations of death. The latter not only signifies an ethical and emotional balancing act for the photographers, but also poses a challenging situation for their mental health (Godulla 2015, p. 7). It is no surprise, then, that the most frequently thematised topic is violence, displayed in 40 of the 62 photos.

In summary, the reality of the World Press Photo of the Year lies in topics of misery and war (news values), is shaped by the visual language the photographers and jury speak (professional actors), and is determined by a Western view of the world. Predicting future decisions made in the World Press Photo of the Year Contest is nearly impossible, but analysing the winners in the further categories—not only the overall World Press Photos of the Year—helps with investigating the specific practices of both the photographers and the juries over the years, and gives even more nuanced insights into the development of the contest and professional photojournalism. Therefore, based on the previously outlined findings, hypotheses are formed in order to be tested in a quantitative manner.

*5.4. Derived Hypotheses for a Subsequent Quantitative Analysis*

5.4.1. Topics

It is assumed that the representation of certain topics is significantly related to the continent where the photo was taken as well as the HDI of the country where it was shot. Following Marfil-Carmona et al. (2018), who identified an anchoring of photography on the topic of sport in the professional elite, it is assumed that sports photos also frequently originate in industrialised nations. Expanding this thought, it is assumed that in general more elite subjects also originate significantly more often in industrialised nations. Further notable topics in line with this assumption are politics, as revealed by the qualitative analysis, as well as science and technology, the latter referring to both the theoretical acquisition of knowledge in research and its practical application. Arts and culture, for example in terms of artistic exhibitions or performing arts, are also assumed to be in line with this. Topics related to nature and the environment, such as landscapes or animals, and religious activities are also expected to correlate with the country where they were shot.

On the other hand, it is assumed that images from continents containing predominantly developing countries, such as Africa, are more strongly characterised by instability (König 2010) and are thus significantly associated with the topic of conflict. This assumption is further supported by the results of the qualitative content analysis of the World Press Photos of the year, which shows a strong presence of the theme of conflict and, at the same time, a dominance of shots from Asia and Africa (see Tables 1 and 4).

**Hypothesis 1 (H1).** *The dominance in the presentation of certain topics depends on the continent of the photographic recording, with continents containing predominantly developing countries (Africa, Asia, South America) correlating more frequently with the topic of conflict, and continents with industrialised nations (Australia/Oceania, Europe, North America) correlating more frequently with the topics of sports, politics, science and technology, arts and culture, nature and environment, and religion.*

As shown in Table 4, with the inclusion of the HDI (from 1990), the World Press Photos of the Year often come from Asian countries, which—in contrast to photographs from countries in Europe or North America—are characterised by a rather low HDI. The same pattern can be seen for winning photos taken in African countries. Taking into account the finding that the majority of winning photos depict conflict, and arguing in line with the

previous assumption of elite topics correlating with industrialised nations, the following hypothesis is put forward:

**Hypothesis 2 (H2).** *There is a significant relationship between a country's HDI and the issues represented in images recorded in that country, with countries that have a lower HDI being predominantly characterised by recordings of conflict while countries with a higher HDI dominate in more elite areas such as sports, politics, science and technology, arts and culture, nature and environment, and religion.*

5.4.2. Negativity

Following Below (2010) and her argument that negative emotions are more likely to generate attention, and Ravel's (2013) finding that images of suffering are predominantly depicted as a result of conflict, it is assumed that negativity correlates in particular with topics that depict crisis situations and attract widespread attention (e.g., conflicts, catastrophes). This assumption is further derived from the results of the qualitative content analysis, which reveal an extremely high proportion of winning photos depicting negativity (62 negative shots vs. five non-negative shots). Taking into account the strong presence of the themes of conflict and catastrophe as well as the high proportion of negative depictions, it is assumed that this correlation is not only evident for the winning photos but for all photos over the decades.

**Hypothesis 3 (H3).** *Depictions of negativity (violence, accident, misery, suffering) correlate particularly strongly with images that address conflicts and catastrophes.*

As König's (2010) study showed, photographs taken in Africa often represent negativity and instability. In addition, Godulla (2009) found a disproportionate representation of winning photographs from crisis areas. With regard to the results of the content analysis (see Tables 2 and 4) that the majority of the winning photos are taken in continents with predominantly developing countries (especially Asia and Africa), and at the same time a high proportion of negative depictions is visible, we assume that this correlation is also evident for all photos in the yearbooks.

**Hypothesis 4 (H4).** *Continents differ significantly in their portrayal of negativity, with continents containing predominantly developing countries (Africa, Asia, South America) being more frequently characterised by negative images than continents containing industrialised nations (Australia/Oceania, Europe, North America).*

Furthermore, in addition to the continental perspective, the level of development of the countries (HDI) from which the recordings originate is also taken into account. As shown in Table 4, photographs in the World Press Photo of the Year often come from Asian or African countries, which are characterised by a rather low HDI. Taking into account the finding that the majority of winning photos depict negativity, the following hypothesis is made:

**Hypothesis 5 (H5).** *The portrayal of negativity correlates with a country's HDI, with countries that have a low HDI being more characterised by negativity in photographs than countries with a very high HDI.*

Finally, the forms of negativity will be examined, subdivided into peril, death, and emotional and physical suffering. In extension to H4, the following assumption will be tested:

**Hypothesis 6 (H6).** *The nature of negativity depicted (peril, death, emotional & physical suffering) differs between continents in that continents with predominantly developing countries (Africa, Asia, South America) are more characterised by frequent depictions of death and suffering than continents with industrialised nations (Australia/Oceania, Europe, North America).*

5.4.3. Power

Following the argumentation of König (2010) that photographs of Africa in the World Press Photo competition are characterised by a Eurocentric view, and the findings of Godulla (2009) that the winning photos often come from crisis areas but their photographers almost always come from Western countries (see also Table 4), a strong dominance of continents with predominantly industrialised nations (especially Europe) and countries with a rather (or very) high HDI is also assumed for the evaluated photos. The assumed correlation not only relates to the number of winners, but also to the number of jury members and participants.

**Hypothesis 7 (H7).** *Europe is significantly over-represented in the number of jury members, participants, and placements compared to continents with predominantly developing countries (Africa, Asia, South America).*

**Hypothesis 8 (H8).** *The higher a country's HDI, the greater its representation in terms of the number of judges, participants, and placements, and the more positively it presents itself and is portrayed by others.*

As Table 4 showed, the winning photos are often taken in crisis areas (especially Asia and Africa), while the photographers mostly originate from Europe or North America. Therefore, it is assumed that this correlation is also prevalent in the photographs of the yearbooks, resulting in an imbalance and thus a power dynamic in the possibility of self and external representation of the individual continents.

**Hypothesis 9 (H9).** *Europe and North America dominate when portraying other continents typified by developing countries (Africa, Asia, South America).*

## 6. Quantitative Content Analysis of World Press Photos

First, with the help of a quantitative content analysis, all photographs ever published in the available World Press Photo yearbooks (divided into six decades) were examined ($N = 11{,}789$). This quantitative method was considered a logical subsequent step after the qualitative analysis, since it usually concerns a high number of messages that are of similar or comparable nature and analyses them with regard to traceable patterns and tendencies (Rössler 2017, p. 17). The results are generated from the systematic analysis of messages (Rössler 2017), in this case, World Press Photos. Compared to the preceding qualitative content analysis, the goal of the quantitative analysis is not to generate a deep understanding of single elements, but a distillation of essential tendencies from the variety of analysed objects (Rössler 2017). Hence, the quantitative content analysis aims to create generalizable statements and to reduce complexity.

The central tendencies were analysed alongside the previously found categories and themes from the qualitative content analysis. They were included in a codebook based on Godulla (2009), which was created in 2019 by two coders. Two pretests were run and showed an intercoder reliability of 1.0 for formal categories, and greater than 0.9 for content categories. The only exceptions were the content categories relating to the topic of the photo and the type of negativity; here, the intercoder reliability was greater than 0.8.

In addition, a further quantitative content analysis of the origin (contents) and Human Development Index of all known jury members ($N = 686$), participants ($N = 132{,}800$) and placements ($N = 2347$) was conducted in order to examine the power structures underlying World Press Photos. Both steps took place between July and December 2020.

The photographs and power structures (jury members, participants, placements) were evaluated in relation to the six decades and the total period, the six continents and the Human Development Index (HDI). The HDI defines four categories of country development (very high: 0.8 and above, high: 0.7–0.8, medium: 0.55–0.7, low: 0.55 and below). In order to analyse the diversity of topics, the photographs were assigned to

the topics identified through the qualitative analysis (conflict, politics, catastrophe, social documentary, society, sports). These six topics were further complemented by the topics of nature & environment, art & culture, science & technology, and religion, in order to assure that every displayed topic could be represented in the analysis. The level of negativity of the photographic presentations is measured with the help of two variables derived from the qualitative study: One distinguishes the negativity of photographs in depicting violence, accident, misery and suffering (or no negativity), the second differentiates photographs according to the degree of suffering (peril, death, emotional suffering, physical suffering).

In order to test the research hypotheses regarding topics and negativity, Chi-Square independence tests and correlation coefficients were used to analyse the correlation of (1) topics and continents/ decades/ HDIs and (2) negativity and continents/ HDIs/ topics using the statistical software SPSS. In a next step, in order to answer the research hypothesis regarding power structures, an ANOVA was conducted to calculate the mean difference in the number of jury members, participants, and placements for the continents and HDIs. In addition, a t-test for independent samples was conducted to test the extent to which the mean values of the HDI differ significantly between the self-presentation and external representation of the continents.

## 7. Results

The evaluation takes place in three steps and according to the formed hypothesis. First, the results are presented with regard to the topics depicted in the World Press Photo competition, followed by an analysis of how negativity is shaped, and finally, an examination of existing power structures.

### 7.1. Perspective 1: Topics Depicted in World Press Photos

Focusing on the variety of topics in the history of World Press Photo, the frequencies of the topics depicted in the photos vary greatly (see Table 5).

**Table 5.** Diversity of topics by continents (1960–2019).

| Topic | | Africa | Asia | Australia/ Oceania | Europe | North America | South America | Total | Chi$^2$(5)/V |
|---|---|---|---|---|---|---|---|---|---|
| Society | yes | **273 (19.6)** | **407 (16.2)** | **43 (20.4)** | **1164 (24.7)** | **425 (18.3)** | **136 (24.8)** | **2448 (21.0)** | 90.7 |
| | no | 1121 (80.4) | 2100 (83.8) | 168 (79.6) | 3540 (75.3) | 1894 (81.7) | 413 (75.2) | 9236 (79.0) | ***/**0.09** |
| Conflict | yes | **388 (27.8)** | **926 (36.9)** | 1 (0.5) | 460 (9.8) | 209 (9.0) | **80 (14.6)** | **2064 (17.7)** | 1106.2 |
| | no | 1006 (72.2) | 1581 (63.1) | 210 (99.5) | 4244 (90.2) | 2110 (91.0) | 469 (85.4) | 9620 (82.3) | ***/**0.31** |
| Sports | yes | 133 (9.5) | 181 7.2) | **100 (47.4)** | 870 (18.5) | 348 (15.0) | 54 (9.8) | **1686 (14.4)** | 391.2 |
| | no | 1261 (90.5) | 2326 (92.8) | 111 (52.6) | 3834 (81.5) | 1971 (85.0) | 495 (90.2) | 9998 (85.6) | ***/**0.18** |
| Politics | yes | 103 (7.4) | 287 (11.4) | 7 (3.3) | 653 (13.9) | **386 (16.6)** | 50 (9.1) | 1486 (12.7) | 100.5 |
| | no | 1291 (92.6) | 2220 (88.6) | 204 (96.7) | 4051 (86.1) | 1933 (83.4) | 499 (90.9) | 10,198 (87.3) | ***/**0.09** |
| Social Document. | Yes | 178 (12.8) | 180 (7.2) | 13 (6.2) | 247 (5.3) | **403 (17.4)** | **88 (16.0)** | 1109 (9.5) | 323.4 |
| | no | 1216 (87.2) | 2327 (92.8) | 198 (93.8) | 4457 (94.7) | 1916 (82.6) | 461 (84.0) | 10,575 (90.5) | ***/**0.17** |
| Nature & Environment | yes | **166 (11.9)** | 143 (5.7) | **21 (10.0)** | 364 (7.7) | 168 (7.2) | 53 (9.7) | 915 (7.8) | 52.8 |
| | no | 1228 (88.1) | 2364 (94.3) | 190 (90.0) | 4340 (92.3) | 2151 (92.8) | 496 (90.3) | 10,769 (92.2) | ***/**0.07** |
| Art & Culture | yes | 57 (4.1) | 54 (2.2) | 6 (2.8) | **386 (8.2)** | 144 (6.2) | 15 (2.7) | 662 (5.7) | 134.4 |
| | no | 1337 (95.9) | 2453 (97.8) | 205 (97.2) | 4318 (91.8) | 2175 (93.8) | 534 (97.3) | 11,022 (94.3) | ***/**0.17** |
| Catastrophe | yes | 58 (4.2) | **212 (8.5)** | **20 (9.5)** | 148 (3.1) | 101 (4.4) | 40 (7.3) | 579 (5.0) | 117.1 ***/**0.1** |
| | no | 1336 (95.8) | 2295 (91.5) | 191 (90.5) | 4556 (96.9) | 2218 (95.6) | 509 (92.7) | 11,105 (95.0) | |
| Science & Technology | yes | 16 (1.1) | 31 (1.2) | 0.0 | **268 (5.7)** | **106 (4.6)** | 13 (2.4) | 434 (3.7) | 136.1 |
| | no | 1378 (98.9) | 2476 (98.8) | 211 (100) | 4436 (94.3) | 2213 (95.4) | 536 (97.6) | 11,250 (96.3) | ***/**0.11** |
| Religion | yes | 22 (1.6) | **86 (3.4)** | 0.0 | 144 (3.1) | 29 (1.3) | **20 (3.6)** | 301 (2.6) | 41.5 |
| | no | 1372 (98.4) | 2421 (96.6) | 211 (100) | 4560 (96.9) | 2290 (98.7) | 529 (96.4) | 11,383 (97.4) | ***/**0.06** |

Note: *** for *p* < 0.001; bold represents significant values.

The subjects most frequently depicted overall are society (21.0%), conflict (17.7%), and sports (14.4%). In contrast, catastrophe (5.0%), science and technology (3.7%), and religion (2.6%) are the least frequently depicted topics. Europe, North America and South America depict social topics more than other topics (24.7%, 18.3%, 24.8%), but this is not a significant correlation. As a chi-square test shows, there is a significant weak correlation between the continents and the topic of sports (Chi2(5) = 391.2, $p < 0.001$, V = 0.18). The topic of sports is particularly dominant in Australia/Oceania (47.4%), while in Asia, for example, only 7.2% of all photos depict sports-related topics. The continents furthermore differ significantly with regard to the depiction of conflict (Chi2(5) = 1106.2, $p < 0.001$), with a medium effect (V = 0.31). Africa and Asia are predominantly characterised by photos portraying conflict (27.8%, 36.9%), while the percentage of conflict photos in Australia/Oceania (0.5%), North America (9.0%) and Europe (9.8%) is significantly lower.

The results further show that the proportion of photos addressing political issues is relatively high, especially for North America (16.6%). With regard to the representation of social documentation by the individual continents, a significant small effect also becomes visible (Chi2(5) = 323.4, $p < 0.001$, V = 0.17), whereby social documentation more frequently originates from the Americas (North America: 17.4%, South America:16.0%) and more rarely from Europe (5.2%) or Australia/Oceania (6.2%).

When looking at photos that deal with nature and the environment, the proportion is relatively higher in Africa (11.9%) and Australia/Oceania (10.0%) than in the other continents. In comparison, the subjects of art and culture are pictured relatively frequently by Europe and North America (8.2%, 6.2%) and relatively infrequently by Asia and South America (2.2%, 2.7%). This is a significant weak relationship (Chi2(5) = 134.4, $p < 0.001$, V = 0.17).

The themes of science and technology are predominantly found in images from Europe and North America (5.7%, 4.6%), while there are hardly any portrayals of this theme from other continents. As the results show, these differences represent a significant weak correlation (Chi2(5) = 136.1, $p < 0.001$, V = 0.11).

The percentage of photos depicting catastrophes is significantly higher for Asia and Australia/Oceania (8.5%, 9.5%) than for all other continents. This is also a significant weak correlation (Chi2(5) = 117.1, $p < 0.001$, V = 0.1). The topic of religion, which is depicted least often overall, is presented proportionally most often by Asia (3.4%) and South America (3.6%). It is noticeable that there are no images from Australia that present religious concerns, and also no presentation of scientific and technical topics by Australia.

While the topics have now been placed into a continental context, Table 6 places the evolution of the subjects into a temporal context for each of the six decades.

As the statistical analysis shows, there is a significant weak correlation in the representation of social topics between the decades (Chi2(5) = 445.7, $p < 0.001$, V = 0.19), with the Sixties and Seventies (1960–1969, 1970–1979) being particularly characterised by these topics (31.5%, 30.1%). In the following three decades, the share of social topics decreases strongly and only in the last decade (2010–2019) can a renewed increase be seen (23.4%).

For the topic of conflict a significant small effect was also found (Chi2(5) = 307.5, $p < 0.001$, V = 0.16). The period from 1990 to 2009 is characterised by a high proportion of conflict recordings (28.1%, 23.8%), whereas the last decade shows a significant decrease in the proportion of conflict recordings (13.9%). The proportion of sports photos is relatively high in the Sixties (1960–1969: 13.6%) and Seventies (1970–1979: 20.5%) and in decade six (2000–2009: 16.1%). While political topics are found more frequently in decades two to four (1960–1989: 21.7%, 13.0%, 20.3%), their share is significantly lower in the following decades with about 6% each. Social documentation is found more frequently for the first time in the period from 1980 to 1989 (10.8%), and after a slight decrease in decade five (1990–1999: 8.4%), becomes particularly frequent in the World Press Photo competition in the last two decades (14.0%, 20.8%).

**Table 6.** Diversity of topics across the decades.

| Topic | | 1960–1969 | 1970–1979 | 1980–1989 | 1990–1999 | 2000–2009 | 2010–2019 | Total | Chi²(5)/V |
|---|---|---|---|---|---|---|---|---|---|
| Society | yes | **511 (31.5)** | **737 (30.1)** | **289 (12.0)** | **300 (15.7)** | 242 (13.7) | **381 (23.4)** | **2.460 (20.9)** | 445.7 |
| | no | 1110 (68.5) | 1713 (69.9) | 2127 (88.0) | 1608 (84.3) | 1526 (86.3) | 1245 (76.6) | 9329 (79.1) | ***/**0.19** |
| Conflict | yes | 199 (12.3) | 278 (11.3) | **404 (16.7)** | **536 (28.1)** | 421 (23.8) | 226 (13.9) | **2.064 (17.5)** | 307.5 |
| | no | 1422 (87.7) | 2172 (88.7) | 2012 (83.3) | 1372 (71.9) | 1347 (76.2) | 1400 (86.1) | 9725 (82.5) | ***/**0.16** |
| Sports | yes | **221 (13.6)** | **503 (20.5)** | 283 (11.7) | **214 (11.2)** | 285 (16.1) | 180 (11.1) | 1.686 (14.3) | 124.8 |
| | no | 1400 (86.4) | 1947 (79.5) | 2133 (88.3) | 1694 (88.8) | 1483 (83.9) | 1446 (89.9) | 10,103 (85.7) | ***/**0.10** |
| Politics | yes | **352 (21.7)** | **318 (13.0)** | **491 (20.3)** | 114 (6.0) | 104 (5.9) | 108 (6.6) | 1.487 (12.6) | 454.0 |
| | no | 1269 (78.3) | 2132 (87.0) | 1925 (79.7) | 1794 (94.0) | 1664 (94.1) | 1518 (93.4) | 10,302 (87.4) | ***/**0.20** |
| Social Document. | Yes | 6 (0.4) | 101 (4.1) | 260 (10.8) | 160 (8.4) | **248 (14.0)** | **339 (20.8)** | 1.114 (9.4) | 535.0 |
| | no | 1615 (99.6) | 2349 (95.9) | 2156 (89.2) | 1748 (91.6) | 1520 (86.0) | 1287 (79.2) | 10,675 (90.6) | ***/**0.21** |
| Nature & Environment | yes | 87 (5.4) | **248 (10.1)** | 167 (6.9) | 129 (6.8) | 154 (8.7) | **190 (11.7)** | 975 (8.3) | 66.1 |
| | no | 1534 (94.6) | 2202 (89.9) | 2249 (93.1) | 1779 (93.2) | 1614 (91.3) | 1436 (88.3) | 10,814 (91.7) | ***/**0.08** |
| Art & Culture | yes | 87 (5.4) | 111 (4.5) | 127 (5.3) | **181 (9.5)** | 142 (8.0) | 16 (1.0) | 664 (5.6) | 1450.4 |
| | no | 1534 (94.6) | 2339 (95.5) | 2289 (94.7) | 1727 (90.5) | 1626 (92.0) | 1610 (99.0) | 11,125 (94.4) | ***/**0.11** |
| Catastrophe | yes | 68 (4.2) | 71 (2.9) | 146 (6.0) | 80 (4.2) | 98 (5.5) | **118 (7.3)** | 581 (4.9) | 52.3 |
| | no | 1553 (95.8) | 2379 (97.1) | 2270 (94.0) | 1828 (95.8) | 1670 (94.5) | 1508 (92.7) | 11,208 (95.1) | ***/**0.07** |
| Science & Technology | yes | 39 (2.4) | 23 (0.9) | 159 (6.6) | **171 (9.0)** | 50 (2.8) | 15 (0.9) | 457 (3.9) | 289.3 |
| | no | 1582 (97.6) | 2427 (99.1) | 2257 (93.4) | 1737 (91.0) | 1718 (97.2) | 1611 (99.1) | 11,332 (96.1) | ***/**0.16** |
| Religion | yes | 51 (3.1) | 60 (2.4) | **90 (3.7)** | 23 (1.2) | 24 (1.4) | 53 (3.3) | 301 (2.6) | 43.1 |
| | no | 1570 (96.9) | 2390 (97.6) | 2326 (96.3) | 1885 (98.8) | 1744 (98.6) | 1573 (96.7) | 11,488 (97.4) | ***/**0.06** |

Note: *** for $p < 0.001$; bold represents significant values.

Finally, we analyse the distribution of topics in terms of a country's HDI. For a better understanding, Table 7 first shows the development of the HDI of the participants' countries from 1990 on. The global development of the HDI is clearly reflected in the proportion of participating countries with a very high or high HDI in recent decades. As shown in Table 7, over the last decade (2000–2019) nearly two-thirds of all participating countries have at least a high HDI.

**Table 7.** The evolution of the Human-Development Index.

| Period | Very High | High | Medium | Low | Unknown | Total |
|---|---|---|---|---|---|---|
| **1990–1999** | 195 (19.9) | 220 (22.4) | **305 (31.1)** | 151 (15.4) | 111 (11.3) | *N* = 982 |
| **2000–2009** | **344 (28.2)** | 302 (24.8) | 303 (24.9) | 210 (17.2) | 60 (4.9) | *N* = 1219 |
| **2010–2019** | **375 (36.9)** | 294 (28.9) | 177 (17.4) | 122 (12.0) | 47 (4.6) | *N* = 1016 |
| **Total Period** | **914 (20.7)** | 816 (18.5) | 785 (17.8) | 483 (10.9) | 218 (4.9) | *N* = 1196 |

Note: Relative frequencies by row; bold represents significant values.

The Tables 8–10 show the distribution of topics with regard to the gradations of the HDI for the last three decades. In the following, we will primarily report conspicuous features and differences for the two contrasting groups (very high/low HDI) and show which topics are the most dominant in percentage terms. The first point to note is that more photos and thus also topics originating from countries with a high or very high HDI are depicted than from countries with a medium or low HDI. This difference is particularly strong from 2010 until 2019, as a total of 67.8 percent of all photos are from countries with at least a high HDI.

**Table 8.** Diversity of topics by a country's HDI (1990–1999).

| Period | Topic | | Very High | High | Medium | Low | Total | Chi²(3)/V |
|---|---|---|---|---|---|---|---|---|
| 1990–1999 | Society | yes | 74 (12.0) | **57 (19.9)** | **95 (24.1)** | 38 (11.6) | **264 (16.2)** | 33.3 ***/**0.15** |
| | | no | 545 (88.0) | 230 (80.1) | 299 (75.9) | 290 (88.4) | 1364 (83.8) | |
| | Conflict | yes | 32 (5.2) | **102 (35.5)** | **114 (28.9)** | **143 (43.6)** | **391 (24.0)** | 215.5 ***/**0.36** |
| | | no | 587 (94.8) | 185 (64.5) | 280 (71.1) | 185 (56.4) | 1237 (76.0) | |
| | Sports | yes | **105 (17.0)** | 26 (9.1) | 38 (9.6) | 32 (9.8) | 201 (12.3) | 19.7 ***/**0.11** |
| | | no | 514 (83.0) | 261 (90.9) | 356 (90.4) | 296 (90.2) | 1427 (87.7) | |
| | Politics | yes | **46 (7.4)** | 10 (3.5) | 27 (6.9) | 12 (3.7) | 95 (5.8) | 9.3 */0.08 |
| | | no | 573 (92.6) | 277 (96.5) | 367 (93.1) | 316 (96.3) | 1533 (94.2) | |
| | Social Documentation | yes | 57 (9.2) | 21 (7.3) | 36 (9.1) | **38 (11.6)** | 152 (9.3) | 3.4/- |
| | | no | 562 (90.8) | 266 (92.7) | 358 (90.9) | 290 (88.4) | 1476 (90.7) | |
| | Nature & Environment | yes | 35 (5.7) | 7 (2.4) | 17 (4.3) | **41 (12.5)** | 100 (6.1) | 32.4 ***/**0.14** |
| | | no | 584 (94.3) | 280 (97.6) | 377 (95.7) | 287 (87.5) | 1528 (93.9) | |
| | Art & Culture | yes | **119 (19.2)** | 28 (9.8) | 26 (6.6) | 7 (2.1) | 180 (11.1) | 77.0 ***/**0.22** |
| | | no | 500 (80.8) | 259 (90.2) | 368 (93.4) | 321 (97.9) | 1448 (88.9) | |
| | Catastrophe | yes | 19 (3.1) | 12 (4.2) | **28 (7.1)** | 9 (2.7) | 68 (4.2) | 12.0 **/0.09 |
| | | no | 600 (96.9) | 275 (95.8) | 366 (92.9) | 319 (97.3) | 1560 (95.8) | |
| | Science & Technology | yes | **122 (19.7)** | 20 (7.0) | 10 (2.5) | 5 (1.5) | 157 (9.6) | 122.0 ***/**0.27** |
| | | no | 497 (80.3) | 267 (93.0) | 384 (97.5) | 323 (98.5) | 1471 (90.4) | |
| | Religion | yes | **10 (1.6)** | 4 (1.4) | 3 (0.8) | 3 (0.9) | 20 (1.2) | 1.8/- |
| | | no | 609 (98.4) | 283 (98.6) | 391 (99.2) | 325 (99.1) | 1608 (98.8) | |
| | **Total** | | **619 (38.0)** | **287 (17.6)** | **394 (24.2)** | **328 (20.2)** | **1628 (100)** | |

Note: * for $p < 0.05$; ** for $p < 0.01$; *** for $p < 0.001$; bold represents significant values.

From 1990 until 1999, sports (17.0%), arts and culture (19.2%), and science and technology (19.7) are particularly dominant for countries with a very high HDI (see Table 8). In addition, countries with a very high HDI have a higher percentage of photographs dealing with politics (7.4%) and religion (1.6%) in this decade compared to countries with a lower HDI. Countries with a low HDI are primarily characterised by conflict issues (43.6%) from 1990 until 1999, while countries with a very high HDI hardly show any depictions of conflict (5.2%).

This low rate is also true for the decade (see Table 9) from 2000 to 2009 (7.6%) and from 2010 to 2019 (3.1%). This difference is a significant mean correlation (Chi2(3) = 215.5, $p < 0.001$, V = 0.36). The huge difference in conflict portrayal between the two HDI levels also marks a significant small effect in the period from 2000 to 2009 (Chi2(3) = 135.0, $p < 0.001$, V = 0.29) and a significant medium effect in the period from 2010 to 2019 (Chi2(3) = 141.5, $p < 0.001$, V = 0.30). In decade five, it can also be observed that conflict issues dominate for countries with a high HDI (35.5%) and medium HDI (28.9%), and likewise in decade six (2000–2009). Furthermore, for countries with a low HDI, it is striking that the share of social documentaries (11.6%) as well as representations of nature and environmental topics (12.5%) exceeds that of the higher HDI levels. For nature and environmental topics, a significant small effect was found (Chi2(3) = 32.4, $p < 0.001$, V = 0.14).

Additionally, countries with a low HDI show a significantly higher proportion of both conflict topics (30.3%) and catastrophes (11.7%) than countries with a high HDI (7.6% and 5.1% respectively). This difference is characterised by a significant weak relationship (Chi2(3) = 39.3, $p < 0.001$, V = 0.15).

**Table 9.** Diversity of topics by a country's HDI (2000–2009).

| Period | Topic | | Very High | High | Medium | Low | Total | Chi²(3)/V |
|---|---|---|---|---|---|---|---|---|
| | Society | yes | 88 (13.9) | **42 (16.8)** | 49 (12.9) | 51 (13.2) | 230 (14.0) | 2.2/- |
| | | no | 543 (86.1) | 208 (83.2) | 332 (87.1) | 335 (86.8) | 1418 (86.0) | |
| | Conflict | yes | 48 (7.6) | **90 (36.0)** | **118 (31.0)** | **117 (30.3)** | 373 (22.6) | 135.0 ***/**0.29** |
| | | no | 583 (92.4) | 160 (64.0) | 263 (69.0) | 269 (69.7) | 1275 (77.4) | |
| | Sports | yes | **160 (25.4)** | 21 (8.4) | 64 (16.8) | 39 (10.1) | 284 (17.2) | 56.7 ***/**0.19** |
| | | no | 471 (74.6) | 229 (91.6) | 317 (83.2) | 347 (89.9) | 1364 (82.8) | |
| | Politics | yes | 36 (5.7) | **19 (7.6)** | 15 (3.9) | 25 (6.5) | 95 (5.8) | 4.3/- |
| | | no | 595 (94.3) | 231 (92.4) | 366 (96.1) | 361 (93.5) | 1553 (94.1) | |
| 2000–2009 | Social Documentation | yes | **119 (18.9)** | 36 (14.4) | 41 (10.8) | 52 (13.5) | 248 (15.0) | 13.5 **/0.09 |
| | | no | 512 (81.1) | 214 (85.6) | 340 (89.2) | 334 (86.5) | 1400 (85.0) | |
| | Nature & Environment | yes | **55 (8.7)** | 20 (8.0) | 24 (6.3) | 23 (6.0) | 122 (7.4) | 3.6/- |
| | | no | 576 (91.3) | 230 (92.0) | 357 (93.7) | 363 (94.0) | 1526 (92.6) | |
| | Art & Culture | yes | **71 (11.3)** | 12 (4.8) | 32 (8.4) | 26 (6.7) | 141 (8.6) | 12.0 **/0.09 |
| | | no | 560 (88.7) | 238 (95.2) | 349 (91.6) | 360 (93.3) | 1507 (91.4) | |
| | Catastrophe | yes | 32 (5.1) | 0 (0.0) | 21 (5.5) | **45 (11.7)** | 98 (5.9) | 39.3 ***/**0.15** |
| | | no | 599 (94.9) | 250 (100) | 360 (94.5) | 341 (88.3) | 1550 (94.1) | |
| | Science & Technology | yes | **22 (3.5)** | 1 (0.4) | 8 (2.1) | 6 (1.6) | 37 (2.2) | 9.2 */0.08 |
| | | no | 609 (96.5) | 249 (99.6) | 373 (97.9) | 380 (98.4) | 1611 (97.8) | |
| | Religion | yes | 0 (0.0) | **9 (3.6)** | 9 (2.4) | 2 (0.5) | 20 (1.2) | 25.4/- |
| | | no | 631 (100) | 241 (96.4) | 372 (97.6) | 384 (99.5) | 1628 (98.8) | |
| | **Total** | | **631 (38.3)** | **250 (15.2)** | **381 (23.1)** | **386 (23.4)** | **1648 (100)** | |

Note: * for $p < 0.05$; ** for $p < 0.01$; *** for $p < 0.001$; bold represents significant values.

In the last decade (2010–2019, see Table 10), countries with a very high HDI again show a significant weak relationship regarding the presentation of sports-related topics compared to countries with a low HDI (16.9% vs. 9.0%, Chi2(3) = 37.0, $p < 0.001$, V = 0.15). Furthermore, countries with a very high HDI are characterised by representations of social issues (25.1%) roughly twice as often as countries with a low HDI (13.2%). This difference can be interpreted as a small effect (Chi2(3) = 20.5, $p < 0.001$, V = 0.12). Countries with a very high HDI also dominate over countries with a low HDI in this decade in photographs that address issues of politics (8.0% vs. 4.3%), arts and culture (1.4% vs. 0.4%), and science and technology (1.4% vs. 0.0%).

In the last decade (2010–2019), countries with a low HDI are again primarily characterised by photographs depicting conflict issues (22.6%). Only countries with a medium HDI depict more conflict during this period (31.6%). This is again a significant medium correlation compared to the proportion of conflict topics from countries with a very high HDI (3.1%). Furthermore, countries with a low HDI also more frequently present social documentation (29.5%) compared to countries with a high or very high HDI (19.2%, 23.5%), and this has more than doubled compared to the previous decade.

**Table 10.** Diversity of topics by a country's HDI (2010–2019).

| Period | Topic | | Very High | High | Medium | Low | Total | Chi²(3)/V |
|---|---|---|---|---|---|---|---|---|
| | Society | yes | 156 (25.1) | **115 (26.6)** | 49 (18.4) | 31 (13.2) | 351 (22.6) | 20.5 ***/**0.12** |
| | | no | 466 (74.9) | 317 (73.4) | 217 (81.6) | 203 (86.8) | 1203 (77.4) | |
| | Conflict | yes | 19 (3.1) | 70 (16.2) | **84 (31.6)** | **53 (22.6)** | 226 (14.5) | 141.5 ***/**0.30** |
| | | no | 603 (96.9) | 362 (83.8) | 182 (68.4) | 181 (77.4) | 1328 (85.5) | |
| | Sports | yes | **105 (16.9)** | 34 (7.9) | 13 (4.9) | 21 (9.0) | 173 (11.1) | 37.0 ***/**0.15** |
| | | no | 517 (83.1) | 398 (92.1) | 253 (95.1) | 213 (91.0) | 1381 (88.9) | |
| | Politics | yes | **50 (8.0)** | 33 (7.6) | 14 (5.3) | 10 (4.3) | 107 (6.9) | 5.3/- |
| | | no | 572 (92.0) | 399 (92.4) | 252 (94.7) | 224 (95.7) | 1447 (93.1) | |
| | Social Documentation | yes | 146 (23.5) | 83 (19.2) | 34 (12.8) | **69 (29.5)** | 332 (21.4) | 23.7 ***/**0.12** |
| | | no | 476 (76.5) | 349 (80.8) | 232 (87.2) | 165 (70.5) | 1222 (78.6) | |
| | Nature & Environment | yes | 61 (9.8) | 40 (9.3) | **48 (18.0)** | 20 (8.5) | 169 (10.9) | 17.3 **/**0.11** |
| | | no | 561 (90.2) | 392 (90.7) | 218 (82.0) | 214 (91.5) | 1385 (89.1) | |
| | Art & Culture | yes | **9 (1.4)** | 5 (1.2) | 1 (0.4) | 1 (0.4) | 16 (1.0) | 3.1/- |
| | | no | 613 (98.6) | 427 (98.8) | 265 (99.6) | 233 (99.6) | 1538 (99.0) | |
| | Catastrophe | yes | 49 (7.9) | 24 (5.6) | 22 (8.3) | **23 (9.8)** | 118 (7.6) | 4.7/- |
| | | no | 573 (92.1) | 408 (94.4) | 244 (91.7) | 211 (90.2) | 1436 (92.4) | |
| | Science & Technology | yes | **9 (1.4)** | 0 (0.0) | 0 (0.0) | 0 (0.0) | 9 (0.6) | 13.6/- |
| | | no | 613 (98.6) | 432 (100) | 266 (100) | 234 (100) | 1545 (99.4) | |
| | Religion | yes | 18 (2.9) | **28 (6.5)** | 1 (0.4) | 6 (2.6) | 53 (3.4) | 20.8 ***/**0.12** |
| | | no | 604 (97.1) | 404 (93.5) | 265 (99.6) | 228 (97.4) | 1501 (96.6) | |
| | **Total** | | **622 (40.0)** | **432 (27.8)** | **266 (17.1)** | **234 (15.1)** | **1554 (100)** | |

The "2010–2019" label spans the Period column.

Note: * for $p < 0.05$; ** for $p < 0.01$; *** for $p < 0.001$; bold represents significant values.

Interim Conclusion I

It emerges that the representation of certain subjects is significantly related to the respective continents from which the recordings originate. First, there is a medium effect between conflict recordings and the continent of their origin. The topic of conflict is more frequently depicted in recordings from Africa, Asia and South America than in recordings from Australia/Oceania, Europe, or North America. Thus, the results for Africa are similar to those of König (2010), who examined portrayals of African populations from 1999 to 2009 and found a strong presence of instability in the recordings considered. Next, sports footage was found to be significantly more likely to come from Australia/Oceania, Europe, and North America than from Africa, Asia, or South America. Thus, following Marfil-Carmona et al. (2018), it appears that sports photographs are not only characterised by photographs of the professional elite, but they are also particularly frequently represented by the elite—in this case, continents with predominantly industrialised nations.

Furthermore, a significant small effect emerges in the depictions of specific subjects, with Europe and North America significantly dominating in the depiction of science and technology-related as well as nature and environment-related subjects compared to all other continents.

From this, the assumption in hypothesis H1 that continents with predominantly developing countries (Africa, Asia, South America) correlate more frequently with the topic of conflict, and continents with industrialised nations (Australia/Oceania, Europe, North America) correlate more frequently with the topics of sports, politics, science and technology, nature and the environment, art and culture, and religion is assumed, taking into account two restrictions. First, the assumed correlation is fully confirmed for the topic conflict and partially for the topics sports, science and technology, and art and culture. Second, it should be qualified that for Australia/Oceania it is the sports topics exclusively

where a significant effect and thus a dominant role is visible, whereas Europe and North America are significantly overrepresented in all three topics (sports, science and technology, and arts and culture) compared to continents with predominantly developing countries.

In addition, significant differences are also evident in terms of thematic representation and HDI of the countries. As can be seen from the results, countries with a low HDI are significantly more likely to have conflict representations in all three decades from 1990 onwards than countries with a very high HDI. In contrast, countries with a very high HDI depict sports significantly more often than countries with a low HDI in all three decades.

Finally, Hypothesis H2, stating that countries with a lower HDI are predominantly characterised by photographs of conflict, while countries with a higher HDI dominate in more elite areas such as sports, politics, and science and technology, is accepted, taking into account two observations. Firstly, the identified correlation relates primarily to conflict and sports, but not (in all decades) to science and technology, nature and environment, arts and culture, and religion. Secondly, political photos are completely excluded from this context.

### 7.2. Perspective 2: Negativity Depicted in World Press Photos

In the following, photographic representations of negativity are analysed in correlation to topics, the continent of the photograph, and the HDI of the country of the photograph. First, Table 11 shows the general development of the depiction of negativity in the World Press Photo competition from 1960 onwards.

**Table 11.** The presentation of negative vs. non-negative photographs.

| Period | Violence | Accident | Misery | Suffering | Total Negativity | No Negativity |
|---|---|---|---|---|---|---|
| **1960–1969** | **265 (16.3)** | 157 (9.7) | 47 (2.9) | 15 (0.9) | 484 (29.9) | **1137 (70.1)** |
| **1970–1979** | **285 (11.6)** | 191 (7.8) | 148 (6.0) | 31 (1.3) | 655 (26.7) | **1795 (73.3)** |
| **1980–1989** | **443 (18.3)** | 183 (7.6) | 223 (9.2) | **199 (8.2)** | 1048 (43.3) | **1368 (56.6)** |
| **1990–1999** | **432 (22.6)** | 89 (4.7) | 208 (10.9) | 109 (5.7) | 838 (43.9) | **1070 (56.1)** |
| **2000–2009** | **428 (24.2)** | 78 (4.4) | 206 (11.7) | 113 (6.4) | 825 (46.7) | **943 (53.3)** |
| **2010–2019** | **366 (22.5)** | 117 (7.2) | 249 (15.3) | 103 (6.3) | **835 (51.4)** | 791 (48.6) |
| **Total Period** | **2219 (18.8)** | 815 (6.9) | 1081 (9.2) | 570 (4.8) | 4685 (39.7) | **7104 (60.3)** |

Note: Relative frequencies by row for each period; bold represents significant values.

In each decade, depictions of violence predominate over depictions of accident, misery and suffering, reaching their highest proportion (24.2%) in decade six (2000–2009). Depictions of misery are the second most common, accounting for a total of 9.2% of all photos submitted. The percentage of photos depicting misery has increased in every decade since 1960. The categories of accident and suffering are the least prevalent (6.9%, 4.8%), with the most photos in which the presentation of suffering is prominent from the decade of 1980 to 1989 (199 photos, 8.2%). Regarding the ratio of negative photos to non-negative photos, it is noticeable that until 2009 the majority of all photos did not show any representations of negativity. However, the proportion of negative photos increased in each decade, so that for the first time in the World Press Photo competition the most recent decade (2010–2019) is characterised by more negative than non-negative photos.

Table 12 depicts the representation of negativity with respect to the themes identified in the photographs. As can be seen from the table, the individual themes correlate to varying degrees with the negativity factors. While themes of conflict and social documentation are particularly likely to have negative depictions (1764, 960), there are very few instances in which the themes of religion and nature and the environment appear in a negative context (55, 18).

Looking at the expressions of negativity, 51% of the social themes and 79.5% of the conflict themes correlate with depictions of violence. In contrast, there are only a few instances in which misery is depicted in the context of social themes (7.8%) and suffering in the context of conflict themes (0.7%). Thereby, a significant weak correlation (Chi2(5) = 124.3,

$p < 0.001$, V = 0.16) can be seen between social topics and the expression of negativity and a particularly strong correlation (Chi2(5) = 1663.7, $p < 0.001$, V = 0.54) with regard to the depiction of conflict topics and the expression of negativity. Furthermore, political topics are particularly often related to depictions of violence (90.5%), while the proportion of accident and misery in political depictions is very low (1.7%, 1.4%). This difference is characterised by a significant weak correlation (Chi2(5) = 355.3, $p < 0.001$, V = 0.28). Lastly, a high proportion of depictions of violence is also apparent for the nature and environment theme (77.8%).

**Table 12.** Correlation of the presentation of negativity and respective topics.

| Topic | | Violence | Accident | Misery | Suffering | Total | Chi$^2$(5)/V |
|---|---|---|---|---|---|---|---|
| Society | yes | **274 (51.0)** | 164 (30.5) | 42 (7.8) | 57 (10.6) | 537 (100) | 124.3 ***/**0.16** |
| | no | 1945 (46.9) | 651 (15.7) | 1039 (25.0) | 513 (12.4) | 4148 (100) | |
| Conflict | yes | **1388 (79.5)** | 15 (0.9) | 330 (18.9) | 13 (0.7) | 1.746 (100) | 1663.7 ***/**0.54** |
| | no | 831 (28.3) | 800 (27.2) | 751 (25.69) | 557 (19.0) | 2939 (100) | |
| Sports | yes | 45 (19.2) | **139 (59.4)** | 23 (9.8) | 27 (11.5) | 234 (100) | 309.8 ***/**0.26** |
| | no | 2174 (48.8) | 676 (15.2) | 1058 (23.8) | 543 (12.2) | 4451 (100) | |
| Politics | yes | **382 (90.5)** | 7 (1.7) | 6 (1.4) | 27 (6.4) | 422 (100) | 355.3 ***/**0.28** |
| | no | 1837 (43.1) | 808 (19.0) | 1075 (25.2) | 543 (12.7) | 4263 (100) | |
| Social Documentation | yes | 90 (9.4) | 11 (1.1) | **575 (59.9)** | 284 (29.6) | 960 (100) | 1561.7 ***/**0.58** |
| | no | 2129 (57.2) | 804 (21.6) | 506 (13.6) | 286 (7.7) | 3725 (100) | |
| Nature & Environment | yes | **14 (77.8)** | 4 (22.2) | 0 | 0 | 18 (100) | - |
| | no | 2205 (47.2) | 811 (17.4) | 1081 (23.2) | 570 (12.2) | 4667 (100) | |
| Art & Culture | yes | 1 (2.7) | 0 | 2 (5.4) | **34 (91.9)** | 37 (100) | 222.1 ***/**0.22** |
| | no | 2218 (47.7) | 815 (17.5) | 1079 (23.2) | 536 (11.5) | 4648 (100) | |
| Catastrophe | yes | 8 (1.5) | **454 (82.8)** | 73 (13.3) | 13 (2.4) | 548 (100) | 1878.8 ***/**0.63** |
| | no | 2211 (5234) | 361 (8.7) | 1008 (24.4) | 557 (13.5) | 4137 (100) | |
| Science & Technology | yes | 0 | 19 (14.8) | 5 (3.9) | **104 (81.3)** | 128 (100) | 600.0 ***/**0.36** |
| | no | 2219 (48.7) | 796 (17.5) | 1076 (23.6) | 466 (10.2) | 4557 (100) | |
| Religion | yes | 17 (30.9) | 2 (3.6) | **25 (45.5)** | 11 (20.0) | 55 (100) | 24.1 ***/**0.07** |
| | no | 2202 (47.6) | 813 (17.6) | 1056 (22.8) | 559 (12.1) | 4630 (100) | |

Note: Relative frequencies by row; * for $p < 0.05$; ** for $p < 0.01$; *** for $p < 0.001$; bold represents significant values.

While sports images are particularly often classified in the category of accident (59.4%), the proportion of misery and suffering in sports images is significantly lower (9.8%, 11.5%). This is a significant small effect (Chi2(5) = 309.8, $p < 0.001$, V = 0.26). Furthermore, portrayals of accidents correlate frequently with the theme of catastrophe (82.8%) and only rarely with the categories of violence (1.5%) or suffering (2.4%). This difference is characterised by a significant large effect (Chi2(5) = 1878.8, $p < 0.001$, V = 0.63).

The negativity category of misery is particularly frequently represented by social documentaries (59.9%) and religious topics (45.5%). In contrast, the proportion of recordings in which the topics of social documentation and religion correlate with the negativity category of accident is very low (1.1%, 3.6%). There is a significant large effect for social documentation and the expression of negativity (Chi2(5) = 1561.7, $p < 0.001$, V = 0.58).

The fourth category of negativity (suffering) is particularly represented by the topics of art and culture (91.9%) and science and technology (81.3%), Whereas there are hardly any cases in which arts and culture or science and technology are depicted in connection with violence, accident or misery.

The differences between continents in the representation of negativity are shown in Table 13. Most negative images come from Asia (30%), followed by Europe (27%). In contrast, there are very few representations of negativity taken in Australia/Oceania (0.1%),

both in each decade and in the overall period. Furthermore, it is noticeable that among all negative photos, the category of violence is the most common (47.4%), dominating in all continents except for Australia/Oceania. The results show that North America dominates in depictions of suffering compared to the other continents (235), and portrayals of misery come particularly often from Africa (305) and Asia (322). Photographs of accidents, on the other hand, predominantly come from Europe (306). For the total period, a significant small effect is observed between the continents and the negativity categories (Chi2(15) = 686.7, *p* < 0.001, V = 0.22).

**Table 13.** Continental differences in the presentation of negativity.

| Period | Category | Africa | Asia | Australia/ Oceania | Europe | North America | South America | Total | Chi$^2$(15)/V |
|---|---|---|---|---|---|---|---|---|---|
| **1960–1969** | Violence | 23 (39.0) | **91 (72.8)** | 0 | 71 (39.0) | **64 (69.6)** | **16 (69.6)** | 265 (55.0) | |
| | Accident | 1 (1.7) | 23 (18.4) | **1 (100)** | 106 (58.2) | 21 (22.8) | 3 (13.0) | 155 (32.2) | - |
| | Misery | **28 (47.5)** | 11 (23.4) | 0 | 2 (1.1) | 2 (2.2) | 4 (17.4) | 47 (9.8) | |
| | Suffering | 7 (11.9) | 0 | 0 | 3 (1.6) | 5 (5.4) | 0 | 15 (3.1) | |
| | **Total** | **59 (12.3)** | **125 (25.9)** | **1 (0.0)** | **182 (37.9)** | **92 (19.1)** | **23 (4.8)** | **482 (100)** | |
| **1970–1979** | Violence | **37 (51.4)** | **89 (47.3)** | 1 (5.3) | **112 (50.5)** | 36 (29.3) | **10 (35.7)** | **285 (43.7)** | |
| | Accident | 15 (20.8) | 8 (4.3) | **18 (94.7)** | 88 (39.6) | **52 (42.3)** | **10 (35.7)** | 191 (29.3) | 206.3 *** |
| | Misery | 14 (19.4) | **89 (47.3)** | 0 | 10 (4.5) | 24 (19.5) | 8 (28.6) | 145 (22.2) | **0.33** |
| | Suffering | 6 (8.3) | 2 (1.1) | 0 | 12 (5.4) | 11 (8.9) | 0 | 31 (4.8) | |
| | **Total** | **72 (11.1)** | **188 (28.7)** | **19 (2.9)** | **222 (34.1)** | **123 (18.9)** | **28 (4.3)** | **652 (100)** | |
| **1980–1989** | Violence | 30 (25.9) | **159 (60.9)** | 0 | 122 (42.5) | 118 (35.1) | 14 (34.1) | **443 (42.3)** | |
| | Accident | 9 (7.8) | 32 (12.3) | 1 (14.3) | 65 (22.6) | 53 (15.8) | **23 (56.1)** | 183 (17.5) | 344.3 *** |
| | Misery | **76 (65.5)** | 59 (22.6) | **6 (85.7)** | 33 (11.5) | 45 (13.4) | 4 (9.8) | 223 (21.3) | **0.33** |
| | Suffering | 1 (0.9) | 11 (4.2) | 0 | 67 (23.3) | **120 (35.7)** | 0 | 199 (19.0) | |
| | **Total** | **116 (11.2)** | **261 (25.0)** | **7 (0.1)** | **287 (27.5)** | **336 (32.2)** | **41 (4.0)** | **1048 (100)** | |
| **1990–1999** | Violence | **82 (48.5)** | **114 (51.4)** | 0 | **187 (67.3)** | 35 (25.2) | **14 (46.7)** | **432 (51.6)** | |
| | Accident | 1 (0.6) | 41 (18.5) | 0 | 24 (8.6) | 22 (15.8) | 1 (3.3) | 89 (10.6) | 135.1 *** |
| | Misery | 72 (42.6) | 43 (19.4) | 0 | 30 (10.8) | **53 (38.1)** | 10 (33.3) | 208 (24.8) | **0.23** |
| | Suffering | 14 (8.3) | 24 (10.8) | 0 | 37 (13.3) | 29 (20.9) | 5 (16.7) | 109 (13.0) | |
| | **Total** | **169 (20.1)** | **222 (26.5)** | **0 (0.0)** | **278 (33.2)** | **139 (16.6)** | **30 (3.6)** | **838 (100)** | |
| **2000–2009** | Violence | 73 (44.8) | 225 (67.2) | 0 | 72 (41.9) | 45 (39.8) | 13 (40.6) | 428 (51.9) | |
| | Accident | 10 (6.1) | 35 (10.4) | 10 (100) | 9 (5.2) | 13 (11.5) | 1 (3.1) | 78 (9.5) | - |
| | Misery | 60 (36.8) | 53 (15.8) | 0 | 59 (34.3) | 16 (14.2) | 18 (56.3) | 206 (25.0) | |
| | Suffering | 20 (12.3) | 22 (6.6) | 0 | 32 (18.6) | 39 (34.5) | 0 | 113 (13.7) | |
| | **Total** | **163 (19.8)** | **335 (40.6)** | **10 (0.1)** | **172 (20.8)** | **113 (13.7)** | **32 (0.4)** | **825 (100)** | |
| **2010–2019** | Violence | **65 (41.4)** | **144 (52.7)** | 0 | 53 (43.4) | 26 (20.8) | 77 (53.8) | 365 (43.9) | |
| | Accident | 8 (5.1) | 49 (17.9) | **10 (83.3)** | 14 (11.5) | 14 (11.2) | 22 (15.4) | 117 (14.1) | 135.9 *** |
| | Misery | 55 (35.0) | 67 (24.5) | 1 (8.3) | 36 (29.5) | **54 (43.2)** | 34 (23.8) | 247 (29.7) | **0.23** |
| | Suffering | 29 (18.5) | 13 (4.8) | 1 (8.3) | 19 (15.6) | 31 (24.8) | 10 (7.0) | 103 (12.4) | |
| | **Total** | **157 (18.9)** | **273 (32.8)** | **12 (0.2)** | **122 (14.7)** | **125 (15.0)** | **143 (17.2)** | **832 (100)** | |
| **Total period** | Violence | 310 (42.1) | 822 (58.5) | 1 (2.0) | 617 (48.9) | 324 (34.9) | 144 (48.5) | 2218 (47.4) | |
| | Accident | 44 (6.0) | 188 (13.4) | **40 (81.6)** | 306 (24.2) | 175 (18.9) | 60 (20.2) | 813 (17.4) | 686.7 *** |
| | Misery | 305 (41.4) | 322 (22.9) | 7 (14.3) | 170 (13.5) | 194 (20.9) | 78 (26.3) | 1076 (23.0) | **0.22** |
| | Suffering | 77 (10.5) | 72 (5.1) | 1 (2.0) | 170 (13.5) | 235 (25.3) | 15 (5.1) | 570 (12.2) | |
| **Total: negative photos** | | **736 (15.7)** | **1404 (30.0)** | **49 (0.1)** | **1263 (27.0)** | **928 (19.8)** | **297 (6.4)** | **4677 (100)** | |

Note: Relative frequencies by column for each period; *** for *p* < 0.001; bold represents significant values.

Although Asia and Europe dominate in the number and proportion of negative images compared to the other continents, Table 14 shows that the ratio of negative to non-negative images varies greatly among these continents. It shows that Asia has the highest proportion, with 56%of all images being negative, but more than half of all images from Africa and South America also depict negativity (52.8%, 54.1%). In comparison, the proportion of negative images from North America is 40% and the proportion of negative images for both Australia/Oceania and Europe is about a quarter (23.2%, 26.6%).

**Table 14.** Relationship of negative to non-negative photographs by continent.

| Negativity | Africa | Asia | Australia/ Oceania | Europe | North America | South America | Total |
|---|---|---|---|---|---|---|---|
| Total: negative photos | 736 (15.7) | **1404 (30.0)** | 49 (0.1) | **1263 (27.0)** | 928 (19.8) | 297 (6.4) | 4677 (100) |
| Total: all photos | 1394 (11.9) | **2507 (21.6)** | 211 (1.8) | **4704 (40.4)** | 2319 (19.5) | 549 (4.8) | 11,684 (100) |
| **Percentage: negative photos/all photos** | **52.8%** | **56.0%** | 23.2% | 26.8% | 40.0% | **54.1%** | 40.0% |

Note: bold represents significant values.

Complementing the continental differences in the representation of negativity, Table 15 maps the individual HDI levels of countries portrayed in the context of images of violence, accident, misery, and suffering. Although countries with a high or very high HDI dominate in overall submission numbers (see Tables 8–10), the majority of depictions of negativity over the total period come from countries with a low HDI (26.3%). Furthermore, for countries with a low HDI, recordings of violence dominate (43.0%), while for countries with a very high HDI, recordings of misery are most common (30.0%). For the total period, a significant weak correlation between expressions of negativity and HDIs is found (Chi2(15) = 267.3, $p < 0.001$, V = 0.20).

**Table 15.** Presentation of Negativity by a country's HDI.

| Period | Category | Very High | High | Medium | Low | Total | Chi$^2$(15)/V |
|---|---|---|---|---|---|---|---|
| **1990–1999** | Violence | 24 (17.1) | **94 (71.2)** | **112 (53.3)** | 69 (35.4) | **299 (44.2)** | |
| | Accident | 23 (16.4) | 10 (7.6) | 23 (11.0) | 21 (10.8) | 77 (11.4) | 137.3 *** |
| | Misery | 45 (32.1) | 10 (7.6) | 50 (23.8) | **88 (45.1)** | 193 (28.5) | |
| | Suffering | **48 (34.3)** | 18 (13.6) | 25 (11.9) | 17 (8.7) | 108 (16.0) | **0.26** |
| | **Total** | 140 (20.7) | 132 (19.5) | **210 (31.0)** | 195 (28.8) | **677 (100)** | |
| **2000–2009** | Violence | **64 (29.6)** | 79 (61.2) | 126 (69.6) | 120 (47.6) | **389 (50.0)** | |
| | Accident | 37 (17.1) | 1 (0.8) | 16 (8.8) | 24 (9.5) | 78 (10.0) | 121.4 *** |
| | Misery | 54 (25.0) | 42 (32.6) | 24 (13.3) | 83 (32.9) | 203 (26.1) | |
| | Suffering | 61 (28.2) | 7 (5.4) | 15 (8.3) | 25 (9.9) | 108 (13.9) | **0.23** |
| | **Total** | 216 (27.8) | 129 (16.6) | 181 (23.2) | **252 (32.4)** | **778 (100)** | |
| **2010–2019** | Violence | 55 (23.8) | **130 (50.2)** | **108 (61.4)** | 68 (45.0) | **361 (44.2)** | |
| | Accident | 49 (21.2) | 20 (7.7) | 24 (13.6) | 24 (15.9) | 117 (14.3) | 100.8 *** |
| | Misery | **77 (33.3)** | 94 (36.3) | 33 (18.8) | 32 (21.2) | 236 (28.9) | |
| | Suffering | 50 (21.6) | 15 (5.8) | 11 (6.3) | 27 (17.9) | 103 (12.6) | **0.20** |
| | **Total** | 231 (28.4) | **259 (31.2)** | 176 (21.7) | 151 (18.7) | **817 (100)** | |
| **Total Period** | Violence | 143 (24.4) | **303 (58.3)** | **346 (61.0)** | 257 (43.0) | **1049 (46.2)** | |
| | Accident | 109 (18.6) | 31 (6.0) | 63 (11.1) | 69 (11.5) | 272 (12.0) | 267.3 *** |
| | Misery | **176 (30.0)** | 146 (18.1) | 107 (18.9) | 203 (33.9) | 632 (27.8) | |
| | Suffering | 159 (27.1) | 40 (7.7) | 51 (9.0) | 69 (11.5) | 319 (14.0) | **0.20** |
| | **Total** | 587 (25.8) | 520 (22.9) | 567 (25.0) | **598 (26.3)** | **2272 (100)** | |

Note: Relative frequencies by column for each level of the HDI and relative frequencies by row for totals of each period; *** for $p < 0.001$; bold represents significant values.

Further differentiation of the visible expression of negativity in the photographs is shown in the Tables 16–18. The analysis is based on the earlier described qualitative content analyses of the categories of negativity, within which both coders independently derived inductive categories for negativity, and found a consensus on four disctinctions through discussion. Negativity differentiates whether:

1.  the lives of people in the photos are potentially or acutely in danger (peril),
2.  abstract, concrete, or massive representations of corpses are present (death),
3.  concrete or massive depictions of emotional suffering are visible (emotional suffering), or
4.  concrete or massive representations of physical suffering are visible (physical suffering).

When analysing the form of negativity in the overall period (see Table 16), representations of concrete emotional suffering (1093) and potential danger to life (1020) dominate across almost all decades. Only in the period from 1980 to 1989 are representations of potential danger to life not among the two most frequently presented forms of negativity, with representations of concrete emotional suffering (233) and massive physical suffering (227) dominating in this decade (see Table 17). Images of corpses, especially abstract (129) and massive depictions (205), are depicted least frequently in the overall period. That depictions of corpses occur less frequently is also apparent in the individual decades.

**Table 16.** Differentiation of the presentation of suffering by continents (1960–2019).

| Period | Category | | Africa | Asia | Australia/ Oceania | Europe | North America | South America | Total |
|---|---|---|---|---|---|---|---|---|---|
| | **Peril** | potential | **156 (70.0)** | **348 (66.5)** | **21 (87.5)** | **304 (67.4)** | **148 (63.5)** | 43 (49.4) | **1020 (66.2)** |
| | | acute | 67 (30.0) | 175 (33.5) | 3 (12.5) | 147 (32.6) | 85 (36.5) | **44 (50.6)** | 521 (33.8) |
| **Total Period** | **Death** | abstract | 15 (15.5) | 31 (15.7) | - | 46 (31.9) | 25 (34.2) | 12 (23.1) | 129 (22.9) |
| | | concrete | 40 (41.2) | 81 (40.9) | - | 56 (38.9) | 30 (41.1) | 23 (44.2) | 230 (40.8) |
| | | massive | 42 (43.3) | **86 (43.4)** | - | 42 (29.2) | 18 (24.7) | 17 (32.7) | 205 (36.3) |
| | **Emotional Suffering** | concrete | **199 (66.8)** | **330 (55.5)** | **7 (100)** | **296 (66.5)** | **216 (67.7)** | **45 (76.3)** | **1093 (63.4)** |
| | | massive | 99 (33.2) | 265 (44.5) | 0 (0.0) | 149 (33.5) | 103 (32.3) | 14 (23.7) | 630 (36.6) |
| | **Physical Suffering** | concrete | 89 (41.6) | 178 (51.9) | 6 (85.7) | 171 (53.1) | 160 (47.9) | 18 (36.0) | 622 (49.0) |
| | | massive | 125 (58.4) | 165 (48.1) | 1 (14.3) | 151 (46.9) | **174 (52.1)** | 32 (64.0) | 648 (51.0) |
| | **Total** | | 832 (16.4) | **1659 (32.7)** | 38 (0.1) | **1362 (26.8)** | 959 (18.9) | 248 (5.0) | **5098 (100)** |

Note: Relative frequencies by row; bold represents significant values.

In terms of continents, most of the visible depictions of negativity come from Asia (32.7%) and Europe (26.8%) and the fewest such images come from Australia/Oceania (0.1%) and South America (5.0%). Furthermore, it is noticeable that most of the depictions of corpses are images from Asia and the number of massive depictions of corpses is particularly high here compared to the other continents (86). In contrast, there are no photographs from Australia/Oceania depicting corpses in any form during the entire period of the World Press Photo competition.

The results in Table 17 show that there are differences in the form of negativity between the continents. In decade two (1960–1969), the proportion of such depictions is highest for images from Europe (32.2%) and Asia (29.5%). The same is true for decade three (1970–1979), in which 33.7% of all negative depictions come from Asia and 29.1% from Europe. Only in the period from 1980 to 1989 is North America most frequently represented, with 29.0% of all such negative recordings, followed by Europe with 27.2% and Asia with 26.5%.

For Africa, there is a clear dominance of depictions of emotional and physical suffering in all three decades, with concrete emotional and massive physical suffering being depicted particularly frequently. Images from Asia are also characterised by depictions of concrete emotional suffering in all three decades. Furthermore, Asia is also dominated in these decades by images in which people are potentially in danger. Thus, the forms of negativity from Asia are similar to those from Europe. In the negative images from Europe, depictions

of concrete emotional suffering and potentially life-threatening situations are frequently found in all the periods presented. For recordings originating from North America, depictions of concrete emotional suffering are equally frequent throughout the period from 1960 to 1989. However, North America differs in the form of negativity portrayed compared to Europe and Asia in that in decade two (1960–1969) concrete physical suffering is depicted second most frequently and in decade four (1980–1989) both forms of physical suffering dominate in the recordings. Only a few images from South America show negativity across the time period. Similar to Europe and Asia, the number of shots of life-threatening situations is slightly higher in South America than the other forms of negativity. Furthermore, decade four (1980–1989) shows an increase in depictions of concrete and massive emotional as well as physical suffering. As mentioned above, there are hardly any negative images for Australia/Oceania in the overall period as well as in the period shown in Table 17.

**Table 17.** Differentiation of the presentation of suffering by continents (1960–1989).

| Period | Category | | Africa | Asia | Australia/ Oceania | Europe | North America | South America | Total |
|---|---|---|---|---|---|---|---|---|---|
| 1960–1969 | Peril | potential | 15 (71.4) | **45 (62.5)** | - | **53 (73.6)** | 18 (48.1) | 3 (27.3) | **134 (64.7)** |
| | | acute | 6 (28.6) | 27 (37.5) | - | 19 (26.4) | 13 (41.9) | **8 (72.7)** | 73 (35.3) |
| | Death | abstract | 0 (0.0) | 0 (0.0) | - | 5 (55.6) | 2 (50.0) | 0 (0.0) | 7 (17.5) |
| | | concrete | 1 (20.0) | 11 (73.3) | - | 3 (33.3) | 2 (50.0) | **6 (85.7)** | 23 (57.5) |
| | | massive | **4 (80.0)** | **4 (26.7)** | - | 1 (11.1) | 0 (0.0) | 1 (14.3) | 10 (25.0) |
| | Emotional Suffering | concrete | **28 (68.3)** | 32 (55.2) | - | 28 (46.7) | **30 (68.2)** | **6 (85.7)** | 124 (59.0) |
| | | massive | 13 (21.7) | 26 (44.8) | - | **32 (53.3)** | 14 (31.8) | 1 (14.3) | 86 (41.0) |
| | Physical Suffering | concrete | 9 (36.0) | 15 (50.0) | - | 31 (62.0) | **21 (75.0)** | 1 (33.3) | 77 (56.6) |
| | | massive | **16 (74.0)** | 15 (50.0) | - | 19 (38.0) | 7 (25.0) | 2 (66.7) | 59 (43.4) |
| | Total | | 92 (15.5) | 175 (29.5) | 0 (0.0) | **191 (32.2)** | 107 (18.0) | 28 (4.7) | **593 (100)** |
| 1970–1979 | Peril | potential | 11 (64.7) | **40 (57.1)** | **12 (100)** | 48 (51.1) | 24 (46.2) | 2 (18.2) | **137 (53.5)** |
| | | acute | 6 (35.3) | 30 (42.9) | 0 (0.0) | 46 (48.9) | 28 (53.8) | **9 (81.8)** | 119 (46.5) |
| | Death | abstract | 0 (0.0) | 0 (0.0) | - | 8 (33.3) | 2 (66.7) | 0 (0.0) | 10 (19.6) |
| | | concrete | 7 (87.5) | 8 (61.5) | - | 10 (41.7) | 1 (33.3) | **3 (100)** | 29 (56.9) |
| | | massive | 1 (12.5) | **5 (38.5)** | - | **6 (25.0)** | 0 (0.0) | 0 (0.0) | 12 (23.5) |
| | Emotional Suffering | concrete | **33 (76.7)** | **56 (60.2)** | - | 41 (74.5) | **30 (62.5)** | 1 (50.0) | **161 (66.8)** |
| | | massive | 10 (23.3) | 37 (39.8) | - | 14 (25.5) | 18 (37.5) | 1 (50.0) | 80 (33.2) |
| | Physical Suffering | concrete | 4 (22.2) | 19 (35.2) | - | 15 (57.7) | 20 (57.1) | 0 (0.0) | 58 (43.0) |
| | | massive | **14 (79.8)** | 35 (64.8) | - | 11 (42.3) | 15 (42.9) | 2 (100) | 77 (57.0) |
| | Total | | 86 (12.6) | **230 (33.7)** | 12 (1.8) | 199 (29.1) | 138 (20.2) | 18 (2.6) | **683 (100)** |
| 1980–1989 | Peril | potential | 18 (64.3) | **48 (62.3)** | 0 (0.0) | **62 (66.7)** | 47 (70.1) | 5 (50.0) | 180 (65.2) |
| | | acute | 10 (35.7) | 29 (37.7) | 1 (100) | 31 (33.3) | 20 (29.9) | 5 (50.0) | 96 (34.8) |
| | Death | abstract | 1 (7.1) | 8 (16.0) | - | 14 (42.5) | 10 (33.3) | 1 (11.1) | 34 (25.0) |
| | | concrete | 7 (50.0) | 20 (40.0) | - | 18 (54.5) | 11 (36.7) | 5 (55.6) | 61 (44.9) |
| | | massive | 6 (42.9) | **22 (44.0)** | - | 1 (3.0) | **9 (30.0)** | 3 (33.3) | 41 (30.1) |
| | Emotional Suffering | concrete | **32 (60.4)** | **64 (55.7)** | 5 (100) | 62 (69.7) | 62 (68.9) | **8 (72.7)** | **233 (64.2)** |
| | | massive | 21 (39.6) | 51 (44.3) | 0 (0.0) | 27 (30.3) | 28 (31.1) | 3 (27.3) | 130 (35.8) |
| | Physical Suffering | concrete | 23 (41.1) | 25 (35.7) | 5 (100) | 50 (47.2) | **67 (43.2)** | 6 (55.5) | 176 (43.7) |
| | | massive | **33 (58.9)** | 45 (64.3) | 0 (0.0) | 56 (52.8) | **88 (56.8)** | 5 (45.5) | **227 (56.3)** |
| | Total | | 151 (12.8) | 312 (26.5) | 11 (0.1) | 321 (27.2) | **342 (29.0)** | 41 (3.4) | **1178 (100)** |

Note: Relative frequencies by row; bold represents significant values.

For the period from 1990 to 1999 (see Table 18), it can again be seen that most of the negative recordings come from Europe (37.4%). The proportion of these depictions from Africa and Asia is also relatively high (24.2%, 23.7%), while no negative images come from Australia/Oceania and only 22 images from South America (2.2%). For the following decades, there is a strong increase in negative depictions originating from Asia (42.3%, 42.2%). Moreover, in the last decade (2010–2019), the percentage of portrayals of negativity from South America increases compared to the previous periods (14.3%). In contrast, increasingly few such images come from Europe, so that the proportion of negative representations from this continent reaches its lowest value so far in the last sample (13.2%).

**Table 18.** Differentiation of the presentation of suffering by continents (1990–a2019).

| Period | Category | | Africa | Asia | Australia/ Oceania | Europe | North America | South America | Total |
|---|---|---|---|---|---|---|---|---|---|
| 1990–1999 | Peril | potential | **46 (75.4)** | **44 (72.1)** | - | **83 (72.8)** | 17 (77.3) | 1 (20.0) | **191 (72.6)** |
| | | acute | 15 (24.6) | 17 (27.9) | - | 31 (27.2) | 5 (22.7) | **4 (80.0)** | 72 (27.4) |
| | Death | abstract | 7 (15.6) | 5 (17.2) | - | 10 (22.3) | 1 (6.7) | 0 (0.0) | 23 (17.0) |
| | | concrete | 18 (40.0) | 10 (34.5) | - | 15 (33.3) | 8 (53.3) | 0 (0.0) | 51 (37.8) |
| | | massive | **20 (44.4)** | 14 (48.3) | - | **20 (44.4)** | 6 (40.0) | 1 (100) | 61 (45.2) |
| | Emotional Suffering | concrete | **36 (50.0)** | **54 (60.0)** | - | **88 (73.3)** | **27 (67.5)** | **7 (78.8)** | **212 (60.6)** |
| | | massive | **36 (50.0)** | 36 (40.0) | - | 51 (36.7) | 13 (32.5) | 2 (22.2) | 138 (39.1) |
| | Physical Suffering | concrete | 20 (35.7) | 28 (56.0) | - | 24 (37.5) | 19 (43.2) | 3 (42.9) | 94 (42.5) |
| | | massive | **36 (64.3)** | 22 (44.0) | - | 40 (62.5) | **25 (56.8)** | **4 (57.1)** | 127 (57.5) |
| | **Total** | | 234 (24.2) | 230 (23.7) | 0 (0.0) | 362 (37.4) | 121 (12.5) | 22 (2.2) | **969 (100)** |
| 2000–2009 | Peril | potential | 23 (56.1) | **61 (55.5)** | **5 (100)** | 25 (67.6) | 20 (69.0) | 3 (37.5) | **137 (59.6)** |
| | | acute | 18 (43.9) | 49 (44.5) | 0 (0.0) | 12 (32.4) | 9 (31.0) | **5 (62.5)** | 93 (40.4) |
| | Death | abstract | 4 (25.0) | 13 (28.9) | - | 6 (25.0) | 6 (50.0) | 0 (0.0) | 29 (28.4) |
| | | concrete | 5 (31.3) | 13 (28.9) | - | 8 (33.3) | 4 (33.3) | 1 (20.0) | 31 (30.4) |
| | | massive | 7 (43.8) | **19 (42.2)** | - | 10 (41.7) | 2 (16.7) | **4 (80.0)** | 42 (41.2) |
| | Emotional Suffering | concrete | **39 (81.3)** | **79 (63.2)** | **1 (100)** | **57 (87.7)** | **36 (76.6)** | 3 (100) | **215 (74.4)** |
| | | massive | 9 (18.7) | 46 (36.8) | 0 (0.0) | 8 (12.3) | 11 (23.4) | 0 (0.0) | 74 (25.6) |
| | Physical Suffering | concrete | **25 (61.0)** | 46 (63.9) | - | **38 (76.0)** | **25 (67.6)** | 1 (100) | 135 (67.2) |
| | | massive | 16 (39.0) | 26 (36.1) | - | 12 (24.0) | 12 (32.4) | 0 (0.0) | 66 (32.8) |
| | **Total** | | 146 (17.8) | **352 (42.3)** | 6 (0.1) | 176 (21.4) | 125 (15.2) | 17 (0.2) | **822 (100)** |
| 2010–2019 | Peril | potential | **43 (78.2)** | **110 (82.7)** | **4 (66.7)** | **33 (80.5)** | 22 (68.8) | **29 (69.0)** | **241 (78.0)** |
| | | acute | 12 (21.8) | 23 (17.3) | **2 (33.3)** | 8 (19.5) | 10 (31.2) | 13 (31.0) | 68 (22.0) |
| | Death | abstract | 3 (33.3) | 5 (10.9) | - | 3 (33.3) | 4 (44.4) | 11 (40.7) | 26 (26.0) |
| | | concrete | 2 (22.2) | 19 (41.3) | - | 2 (22.2) | 4 (44.4) | 8 (29.6) | 35 (35.0) |
| | | massive | 4 (44.4) | **22 (47.8)** | - | 4 (44.4) | 1 (11.2) | 8 (29.6) | 39 (39.0) |
| | Emotional Suffering | concrete | **31 (75.6)** | 45 (39.5) | 1 (100) | **20 (54.1)** | 31 (62.0) | **20 (74.1)** | **148 (54.8)** |
| | | massive | 10 (24.4) | **69 (60.5)** | 0 (0.0) | 17 (45.9) | 19 (38.0) | 7 (25.9) | 122 (45.2) |
| | Physical Suffering | concrete | 8 (44.4) | 45 (67.2) | 1 (50.0) | 13 (50.0) | 8 (22.9) | 7 (26.9) | 82 (47.1) |
| | | massive | 10 (55.6) | 22 (32.8) | 1 (50.0) | 13 (50.0) | **27 (77.1)** | 19 (73.1) | 92 (52.9) |
| | **Total** | | 123 (14.4) | **360 (42.2)** | 9 (0.1) | 113 (13.2) | 126 (14.8) | 122 (14.3) | **853 (100)** |

Note: Relative frequencies by row; bold represents significant values.

Regarding the different categories of negativity, a large proportion of the recordings show a visible manifestation of concrete emotional suffering. This is true for Africa and Asia as well as Europe and North America. In Africa, decades five (1990–1999) and seven (2010–2019) are also dominated by images in which the people depicted are in potential mortal danger. In the period from 2000 to 2009, specific images of physical suffering are frequently found alongside depictions of emotional suffering.

Across the time periods, images originating from Asia are also frequently characterised by concrete depictions of potentially life-threatening situations as well as physical suffering. Thus, the proportion of such images is similar to that of images from Europe, where the visible framing of negativity is dominated by images of concrete emotional suffering as well as potential life-threatening situations in the period 1990–2019. In decade six (2000–2009), it is also noticeable that there are high levels of portrayal of concrete physical suffering from Europe.

### 7.2.1. Depictions of Death

In terms of visible depictions of death, there is a change over time from predominantly concrete depictions of death in decades two to four (1960–1989) to predominantly massive depictions of death from 1990 onwards. Decades four and five (1980–1989, 1990–1999) have the highest number of images of death (136, 135). The overall period is characterised by concrete portrayals of this theme (230) followed by massive depictions (205). In contrast, there are fewer abstract depictions of death (129). Furthermore, in the overall period most of the images in this category come from Asia (198), Europe (144) and Africa (97), while there are fewer such images coming from North America (73) and South America (52) and not a single image coming from Australia/Oceania. The continents differ not only in the number of depictions of death, but also in their degree of abstraction. Africa with 43.3% and Asia with 43.4% of all recordings of death are most characterised by massive representations, while for recordings from Europe, North America and South America concrete representations of death are proportionally the most common (38.9%, 41.1%, 44.2%).

### 7.2.2. Interim Conclusion II

The results of the quantitative analysis show a strong increase in negative images compared to non-negative images; a trend particularly prominent in the period starting in 2010. In this decade, the majority of the evaluated photographs show the representation of at least one negativity category.

When focusing on the correlation between the subjects depicted and the categories of negativity, significant large effects are found for photographs of conflict and catastrophe. The results are similar to those of Ravel (2013), in that shots of suffering are often depicted in the context of conflict. Moreover, it was found that not only is the proportion of winning photographs depicting negativity and conflict and catastrophe very high, but also the proportion of photographs from each yearbook. Therefore, Hypothesis H3, which asserts that depictions of negativity (violence, accident, misery, suffering) correlate particularly with photographs depicting conflict and catastrophe, is accepted. It should be noted that conflicts most often depict forms of violence, while catastrophes most often depict accidents. Furthermore, significant large effects were also apparent for the themes of social documentation and science and technology, with social documentation in particular frequently depicting misery and the theme of science and technology most frequently depicting suffering.

The results of the continental perspective on depictions of negativity tie in with the findings of König (2010) that footage from Africa frequently depicts negativity and instability. Similarly, parallels emerge from evaluations of the winning photographs, with the finding that the majority of winning photographs are from continents with predominantly developing countries, while at the same time a high proportion of negativity depictions are visible overall. Looking at the individual decades, it can be seen that Europe plays

a dominant role in the depiction of negativity in the period up to 1999. From 2000 onwards, the number of negative images from Asia exceeds that of Europe. Across the total period Asia and Europe provide the most representations of negativity, while there are significantly fewer such images from South America and Australia/Oceania. For several decades as well as the overall period, the continents differ significantly in the number and type of representations of negativity. While the categories of violence and misery are most pronounced in Africa, Asia and South America, depictions of violence are particularly common in Europe and North America. Australia, on the other hand, is characterised almost exclusively by images of accidents.

As was further shown, it is useful to consider the number of negative images in relation to the total number of images for the individual continents. Although Europe, along with Asia, produces the most negative depictions, it also produces the most non-negative depictions unlike Asia, which is much more dominated by negativity. Furthermore, Africa and South America are also proportionally more dominated by negative portrayals than North America and Australia/Oceania. For this reason, Hypothesis H4 is accepted: Continents differ significantly in their portrayal of negativity, with continents with predominantly developing countries (Africa, Asia, South America) being more frequently characterised by negative images than continents with industrialised nations (Australia/Oceania, Europe, North America).

From the perspective of the development level of the countries (HDI) from which negative images originate, a dominant role of countries with a low HDI becomes apparent. Although the proportion of participating countries that have a low HDI is significantly lower than that of countries with a high or very high HDI, most negative photographs were shot in these countries. While negativity in countries with a very high HDI overall is most often depicted in the form of misery, countries with an lower HDI are most characterised by depictions of violence. Given the (albeit only slightly) higher total number of negatives from countries with a low HDI, Hypothesis H5 is accepted: The portrayal of negativity depends on a country's HDI, with countries with a low HDI being more characterised by negativity in photographs than countries with a very high HDI.

Finally, we investigated whether there is a correlation between the negativity categories peril, death, and emotional and physical suffering and the continents of the photographs. Similar to the general categories of negativity (violence, accident, misery, and suffering), until 1999 these are particularly strongly dominated by recordings from Europe, while from 2000 and in the overall period, recordings from Asia strongly dominate. Fewer massive depictions of death and suffering, on the other hand, come from South America and Australia/Oceania.

Hypothesis H6, which proposes that the form of negativity (peril, death, emotional and physical suffering) differs between continents, with continents with predominantly developing countries (Africa, Asia, South America) being characterised more frequently by massive depictions of death and suffering than continents with predominantly industrialised nations (Australia/Oceania, Europe, North America), can only be answered in part. The assumed correlation applies primarily to massive depictions of death and can be justified on the basis of the higher number of such images from Africa and Asia compared to Europe and North America. While massive depictions of emotional suffering are also most frequently from Asia, this category is also frequently characterised in images from Europe, North America, and Africa. The same is true for depictions of massive physical suffering. This category is most heavily characterised by footage from North America, followed by footage from Asia, Europe, and Africa.

### 7.3. Perspective 3: Power Structures Displayed through World Press Photos

In the following, the developments in the number of jury members, participants and placements in the World Press Photo competition are examined with regard to existing power structures. In doing so, consideration is given to the origin (continents) of the

participating individuals, the HDI of the country from which they originate, as well as the number of placements and the possibility of self and other representations of the continents.

Table 19 shows the over- and under-representation of continents in the number of judges, participants, and placements in each decade as well as overall. First, the number of jury members has increased in each period, and the number of participants as well as the number of prizes (except for the last period) have also increased sharply. Since the beginning of the World Press Photo competition, a total of 686 jury members have judged photos submitted by 132,799 participants and awarded placements to 2347 of these submissions.

**Table 19.** Over- and under-representation of jury members, participants, and placements by continent.[4]

| Period | Category | Africa | Asia | Australia/ Oceania | Europe | North America | South America | Total |
|---|---|---|---|---|---|---|---|---|
| **Until 1959** | **Jury** | - | - | - | **28 (93.3)**[5] | **2 (6.7)** | - | 30 (100) |
| | Participants | - | - | - | - | - | - | - |
| | Rankings | - | - | - | - | - | - | - |
| **1960–1969** | **Jury** | - | 1 (0.02) | - | **71 (87.7)** | **9 (0.11)** | - | 81 (100) |
| | Participants | 28 (2.8) | 65 (6.5) | 15 (1.5) | **704 (70.0)** | **174 (17.3)** | 19 (1.9) | 1005 (100) |
| | Rankings | - | - | - | - | [6]- | - | - |
| **1970–1979** | **Jury** | 1 (1.3) | 4 (5.1) | 1 (1.3) | **62 (79.5)** | **9 (11.5)** | 1 (1.3) | 78 (100) |
| | Participants | 261 (4.6) | 469 (8.2) | 201 (3.5) | **3517 (61.8)** | **1151 (20.2)** | 88 (1.5) | 5687 (100) |
| | Rankings | 7 (2.7) | 11 (4.3) | 7 (2.7) | **164 (64.3)** | **65 (25.5)** | 1 (0.4) | 255 (100) |
| **1980–1989** | **Jury** | 1 (1.1) | 6 (6.7) | - | **71 (78.9)** | **11 (12.2)** | 1 (1.1) | 90 (100) |
| | Participants | 273 (2.8) | 1168 (12.1) | 174 (1.8) | **5461 (56.4)** | **2444 (25.2)** | 164 (1.7) | 9684 (100) |
| | Rankings | 2 (0.4) | 16 (3.3) | 2 (0.4) | **252 (51.6)** | **210 (43.0)** | 6 (1.2) | 488 (100) |
| **1990–1999** | **Jury** | 6 (6.1) | **16 (16.3)** | - | **55 (56.1)** | 13 (13.3) | 8 (8.2) | 98 (100) |
| | Participants | 849 (3.0) | 5526 (19.4) | 592 (2.1) | **13,247 (46.5)** | 6526 (22.9) | 1759 (6.2) | 28,499 (100) |
| | Rankings | 21 (3.7) | 21 (3.7) | 20 (3.5) | **292 (51.4)** | 201 (35.4) | 12 (2.1) | 567 (100) |
| **2000–2009** | **Jury** | 14 (10.1) | 17 (12.2) | 6 (4.3) | **68 (48.9)** | **26 (18.7)** | 8 (5.8) | 139 (100) |
| | Participants | 1334 (2.9) | **10,478 (22.9)** | 1313 (2.9) | **20,157 (44.0)** | 9430 (20.6) | 3122 (6.8) | 45,834 (100) |
| | Rankings | 23 (3.7) | 61 (9.7) | 35 (5.6) | **319 (50.6)** | **163 (26.0)** | 26 (4.1) | 627 (100) |
| **2010–2019** | **Jury** | 12 (7.1) | 26 (15.3) | 3 (1.8) | **82 (48.2)** | **33 (19.4)** | 14 (8.2) | 170 (100) |
| | Participants | 1134 (2.7) | **11,384 (27.0)** | 816 (1.9) | **19,933 (47.7)** | 6157 (14.6) | 2666 (6.3) | 42,090 (100) |
| | Rankings | 18 (4.4) | 57 (13.9) | 22 (5.4) | **213 (52.0)** | **80 (19.5)** | 20 (4.9) | 410 (100) |
| **Total Period** | **Jury** | 34 (5.0) | 70 (10.2) | 10 (1.5) | **437 (63.7)** | **103 (15.0)** | 32 (4.7) | 686 (100) |
| | Participants | 3879 (2.9) | **29,090 (21.9)** | 3111 (2.3) | **63,019 (47.5)** | 25,882 (19.5) | 7818 (5.9) | 132,799 (100) |
| | Rankings | 71 (3.0) | 166 (7.1) | 86 (3.7) | **1240 (52.8)** | **719 (30.6)** | 65 (2.8) | 2347 (100) |

Note: Relative frequencies by row; bold represents significant values.

At this point, it should be noted that the distribution of jury members, participants and placements, however, differs greatly between continents. As the figures for the two continents with the highest shares of the individual categories make clear, Europe and North America are strongly overrepresented in the number of jury members, participants and placements. As the results show, nearly two-thirds of all jury members (63.7%) and half of all participants (47.5%) over the total period are from Europe, and more than half of all placements (52.8%) were taken by European entries. This means no other continent has been as successful as Europe. North America stands out in second place with 15% of all jury members and 30.6% of all placements. On the other hand, the most participants after Europe come from Asia (21.9%), although submissions from this continent achieve significantly fewer placements than Europe and North America. The dominance of Europe and North America in the power structures laid out is evident in almost all categories of the individual decades. In general, the number of jury members and participants from

Asia has greatly increased over the past two decades, and more submissions from Asia have been awarded placements.

While Europe, North America, and to some extent Asia dominate the categories shown, the other continents are severely underrepresented. As the data show, the percentage of judges from Africa (5.0%), Australia/Oceania (1.5%) and South America (4.7%) is significantly lower than that of Asia, Europe and North America. Only 2.9% of all participants come from Africa, 2.3% from Australia/Oceania and 5.9% from South America, and thus the proportion of participants from these continents is also significantly lower than that of Asia, Europe and North America. The fewest placements overall were awarded to submissions from South America (65, 2.8%) and Africa (71, 3.0%). Although Australia/Oceania has the lowest number of entrants in the overall period, this continent is more successful than Africa and Asia in terms of number of placements, with 86 (3.7%).

Table 20 shows the average differences in the number of jury members, participants and placements based on the average values of these categories for Europe. As the conducted ANOVAs show, there are significant differences between the average number of jury members, participants and placements between the continents, with Europe dominating over the majority of other continents in these categories.

**Table 20.** Mean difference in the number of jury members, participants and placements.

| Period | Category | Europe | Africa | Asia | Australia/ Oceania | North America | South America | ANOVA |
|---|---|---|---|---|---|---|---|---|
| **1960–1969** | **Jury** | 0.54 | **0.54 \*\*\*** | **0.51 \*\*\*** | **0.54 \*\*\*** | −0.11 | **0.54 \*\*\*** | $F_{(5,210)}$ = 4.51 \*\* |
| | **Participants** | 5.33 | **3.67 \*\*\*** | **3.30 \*\*\*** | **3.19 \*\*** | −7.10 | **3.98 \*\*\*** | $F_{(5,210)}$ = 6.71 \*\*\* |
| | **Rankings** | - | - | - | - | - | - | - |
| **1970–1979** | **Jury** | 0.31 | **0.28 \*\*\*** | **0.26 \*\*\*** | **0.25 \*** | 0.01 | **0.28 \*\*\*** | $F_{(5,395)}$ = 7.56 \*\*\* |
| | **Participants** | 17.59 | **10.53 \*\*\*** | **12.13 \*\*\*** | 5.02 | −20.78 | **14.84 \*\*\*** | $F_{(5,395)}$ = 11.81 \*\*\* |
| | **Rankings** | 0.82 | **0.63 \*\*\*** | **0.69 \*\*\*** | 0.38 | −1.35 | **0.79 \*\*\*** | $F_{(5,395)}$ = 10.29 \*\*\* |
| **1980–1989** | **Jury** | 0.28 | **0.26 \*\*\*** | **0.24 \*\*\*** | **0.28 \*\*\*** | 0.04 | **0.27 \*\*\*** | $F_{(5,547)}$ = 13.51 \*\*\* |
| | **Participants** | 21.84 | **15.78 \*\*\*** | **13.56 \*\*\*** | **12.18 \*\*** | −32.47 | **18.81 \*\*\*** | $F_{(5,547)}$ = 14.75 \*\*\* |
| | **Rankings** | 1.01 | **0.96 \*\*\*** | **0.90 \*\*\*** | **0.90 \*\*\*** | −3.66 | **0.90 \*\*\*** | $F_{(5,547)}$ = 17.43 \*\*\* |
| **1990–1999** | **Jury** | 0.16 | **0.12 \*\*\*** | **0.09 \*\*** | **0.16 \*\*\*** | 0.04 | 0.07 | $F_{(5,976)}$ = 4.71 \*\*\* |
| | **Participants** | 37.42 | **31.76 \*\*\*** | **15.75 \*\*\*** | 7.82 | −20.85 | **18.09 \*\*\*** | $F_{(5,976)}$ = 10.18 \*\*\* |
| | **Rankings** | 0.82 | **0.69 \*\*\*** | **0.74 \*\*\*** | −0.18 | −0.97 | **0.69 \*\*\*** | $F_{(5,976)}$ = 11.56 \*\*\* |
| **2000–2009** | **Jury** | 0.18 | **0.11 \*\*** | **0.13 \*\*\*** | −0.12 | −0.01 | **0.10 \*** | $F_{(5,1213)}$ = 6.31 \*\*\* |
| | **Participants** | 52.09 | **46.16 \*\*\*** | **21.09 \*\*** | −13.57 | −16.25 | **23.96 \*\*\*** | $F_{(5,1213)}$ = 13.46 \*\*\* |
| | **Rankings** | 0.82 | **0.72 \*\*\*** | **0.64 \*\*\*** | −0.93 | −0.36 | **0.59 \*\*\*** | $F_{(5,1213)}$ = 16.34 \*\*\* |
| **2010–2019** | **Jury** | 0.25 | **0.18 \*\*\*** | **0.16 \*\*** | 0.07 | −0.05 | 0.09 | $F_{(5,1008)}$ = 4.63 \*\*\* |
| | **Participants** | 60.77 | **54.47 \*\*\*** | 20.11 | 9.77 | 5.80 | **30.13 \*\*\*** | $F_{(5,997)}$ = 8.86 \*\*\* |
| | **Rankings** | 0.64 | **0.54 \*\*\*** | **0.44 \*\*\*** | −0.65 | −0.07 | **0.42 \*\*\*** | $F_{(5,1008)}$ = 12.19 \*\*\* |
| **Total Period** | **Jury** | 0.26 | **0.21 \*\*\*** | **0.19 \*\*\*** | **0.16 \*\*\*** | 0.03 | **0.18 \*\*\*** | $F_{(5,4.404)}$ = 37.92 \*\*\* |
| | **Participants** | 37.53 | **31.61 \*\*\*** | **11.89 \*\*\*** | 5.79 | −19.60 | **17.54 \*\*\*** | $F_{(5,4.404)}$ = 32.92 \*\*\* |
| | **Rankings** | 0.74 | **0.63 \*\*\*** | **0.59 \*\*\*** | −0.14 | **−0.85 \*\*** | **0.57 \*\*\*** | $F_{(5,4.404)}$ = 50.40 \*\*\* |

Note: The mean differences are based on the mean value for Europe. \* for $p < 0.05$; \*\* for $p < 0.01$; \*\*\* for $p < 0.001$; bold represents significant values.

For the entire period, we first find a highly significant difference between the average number of jury members and their origin (continent) ($F_{(5,4.404)}$ = 37.92, $p < 0.001$). These significant differences are also present in the individual decades. Countries from Europe participating in the World Press Photo competition have a significantly higher average number of jury members than countries from Africa, Asia, Australia/Oceania or South America participating in the competition (each $p < 0.001$). Next, it is visible overall that there are also highly significant differences between continents in terms of the average

number of participants (F(5,4.404) = 32.92, *p* < 0.001). Countries from Europe participating in the competition have an average of around 38 participants, a significantly higher average number of participants than countries from Africa, Asia or South America participating in the competition (*p* < 0.001 in each case). Last, the overall period also shows significant differences between continents in the average number of placements (F(5,4.404) = 50.40, *p* < 0.001). As can be seen from ANOVA, a European country participating in the competition has a significantly higher average number of placements than a country from Africa, Asia or South America participating in the competition (each *p* < 0.001). At the same time, it is noticeable that the average number of placements for a European country participating in the competition is significantly lower than the average number of placements for a participating country from North America (*p* < 0.01).

Regarding the significant average differences in the number of placements from Europe compared to Africa, Asia and South America, this increased until 1989 and decreased from the following decade. For example, if a European country participated in the competition a total of 100 times from 1980 to 1989, a total of about 101 placements would go to Europe in that period, while the same number of participations by an African country would result in in only five placements, around 96 fewer. Although the average difference in the number of placements between Europe and Africa decreased significantly by the last decade, it is still characterised by a highly significant relationship (*p* < 0.001). If a European country participated in the competition a total of 100 times in the period from 2010 to 2019, it would achieve a total of around 64 placements, while the same number of participations by an African country would result in only ten placements, a difference of 54.

Finally, the significant differences in these power structures relate primarily to the comparison of Europe with Africa, Asia and South America. With regard to Australia/Oceania, a dominant European role can be seen primarily in the average number of jury members until the end of the 1990s as well as for the overall period. Regarding the average number of jury members, participants and placements from North America, only one significant difference can be observed compared to Europe. This relates to the average number of placements over the total period, which is significantly lower for Europe than for North America.

Based on the continental differences in the average number of jury members, participants and placements, the most successful countries of the individual continents are shown in Table 21. As can be seen from the table, France and the USA in particular are characterised by a high number of placements and a high ratio of placements per 100 participants over the entire period of the World Press Photo competition. For every 100 participants from France, there are 3.58 placements and for every 100 participants from the USA, 3.17 placements. It is striking that the three most successful African, Asian and South American countries together have fewer placements than either the most successful European or North American country (France, USA).

The dominance of Europe and North America (primarily the USA) also becomes clear when comparing the ratio of placements per 100 participants of Asian and South American countries. While China, for example, also has a very high number of participants, the country achieved a total of only 42 placements, equating to only 0.4 placements per 100 participants. Likewise, Brazil—the country with the most placements and participants in South America—has only 0.7 placements per 100 participants and is thus also well behind the leaders from Europe and North America.

**Table 21.** Front runner in the number of placements and participants (1960–2019).

| Continent | Country | Rankings | Participants | Score |
|---|---|---|---|---|
| | **South Africa** | **56** | **1827** | **3.07** |
| **Africa** | Egypt | 3 | 328 | 0.91 |
| | Algeria | 3 | 76 | 3.95 |
| | China | 42 | 10,419 | 0.40 |
| **Asia** | India | 19 | 4016 | 0.47 |
| | Israel | 17 | 1114 | 1.53 |
| **Australia/Oceania** | Australia | 82 | 2577 | 3.18 |
| | New Zealand | 4 | 533 | 0.75 |
| | **France** | **215** | **6002** | **3.58** |
| **Europe** | Great Britain | 164 | 7747 | 2.12 |
| | Italy | 107 | 6636 | 1.61 |
| | **USA** | **659** | **20,770** | **3.17** |
| **North America** | Canada | 38 | 2453 | 1.55 |
| | Mexico | 15 | 1474 | 1.02 |
| | Brazil | 22 | 3061 | 0.72 |
| **South America** | Argentina | 16 | 1556 | 1.03 |
| | Columbia | 8 | 937 | 0.85 |

Note: The score is based on the number of rankings per 100 participants; bold represents significant values.

Looking at countries from Africa, it is noticeable that South Africa is characterised by success at the competition. Even though the country has a low total number of placements and participants compared to the listed European countries, it still shows a high ratio of placements per 100 participants (3.07). In contrast, the next most successful African countries (Egypt, Algeria) are characterised by an extremely lower number of placements and participants. This makes South Africa the only country on the continent to successfully participate in the World Press Photo competition. Lastly, in Australia/Oceania, Australia itself is the most successful country in the competition. Like South Africa, it has a lower total number of placements and participants, but a high score of placements per 100 participants (3.18). Following on from the consideration of geographical differences and success in the World Press Photo competition, the HDI of the country of entry is used in Table 22 to analyse existing power structures.

**Table 22.** Mean difference in the number of jury members, participants and placements by HDI.

| Period | Category | Very High | High | Medium | Low | Unknown | ANOVA |
|---|---|---|---|---|---|---|---|
| | Jury | 0.27 | **0.18 \*\*\*** | **0.22 \*\*\*** | **0.21 \*\*\*** | **0.24 \*\*\*** | $F_{(4,977)} = 20.01$ \*\*\* |
| 1990–1999 | Participants | 80.24 | **59.41 \*\*\*** | **63.18 \*\*\*** | **64.60 \*\*\*** | **73.87 \*\*\*** | $F_{(4,977)} = 41.12$ \*\*\* |
| | Rankings | 2.18 | **1.78 \*\*\*** | **2.07 \*\*\*** | **2.15 \*\*\*** | **2.09 \*\*\*** | $F_{(4,977)} = 33.34$ \*\*\* |
| | Jury | 0.28 | **0.23 \*\*\*** | **0.23 \*\*\*** | **0.24 \*\*\*** | **0.28 \*\*\*** | $F_{(4,1.214)} = 22.19$ \*\*\* |
| 2000–2009 | Participants | 82.03 | **60.49 \*\*\*** | **53.79 \*\*\*** | **71.81 \*\*\*** | **75.23 \*\*\*** | $F_{(4,1.214)} = 37.42$ \*\*\* |
| | Rankings | 1.47 | **1.33 \*\*\*** | **1.26 \*\*\*** | **1.42 \*\*\*** | **1.40 \*\*\*** | $F_{(4,1.214)} = 45.56$ \*\*\* |
| | Jury | 0.34 | **0.25 \*\*\*** | **0.29 \*\*\*** | **0.28 \*\*\*** | **0.34 \*\*\*** | $F_{(4,1.010)} = 15.11$ \*\*\* |
| 2010–2019 | Participants | 71.35 | **35.00 \*\*\*** | **49.39 \*\*\*** | **66.56 \*\*\*** | **67.74 \*\*\*** | $F_{(4,1.010)} = 19.65$ \*\*\* |
| | Rankings | 0.81 | **0.56 \*\*\*** | **0.67 \*\*\*** | **0.73 \*\*\*** | **0.79 \*\*\*** | $F_{(4,1.010)} = 21.43$ \*\*\* |
| | Jury | 0.3 | **0.23 \*\*\*** | **0.25 \*\*\*** | **0.25 \*\*\*** | **0.29 \*\*\*** | $F_{(4,3.211)} = 55.74$ \*\*\* |
| Total Period | Participants | 77.27 | **50.58 \*\*\*** | **54.79 \*\*\*** | **66.72 \*\*\*** | **71.37 \*\*\*** | $F_{(4,3.211)} = 90.15$ \*\*\* |
| | Rankings | 1.35 | **1.10 \*\*\*** | **1.20 \*\*\*** | **1.30 \*\*\*** | **1.28 \*\*\*** | $F_{(4,3.211)} = 82.90$ \*\*\* |

Note: The mean differences are based on the mean value for countries with a very high HDI. \*\*\* for $p < 0.001$; bold represents significant values.

For the total period, a significant difference between the average number of jury members and the HDI level of the participating countries is apparent ($F(4,1.011) = 55.74$, $p < 0.001$). Countries with a very high HDI participating in the competition have on average a higher number of jury members than countries with a lower HDI level. This significant difference is observed in all time periods, with the greatest average difference between countries with a very high HDI and countries with a low HDI in all decades. For example, if a country with a very high HDI participated in the competition a total of 100 times in the period from 2010 to 2019, a total of between three and four jury members would come from a country with this HDI level in this period, while the same number of participations from a country with a low HDI would result in almost three fewer jury members, and thus on average not even one jury member coming from a country with a low HDI.

Furthermore, significant differences between the average number of participants and the HDIs of the participating countries are evident over the entire period ($F(4,3.211) = 90.15$, $p < 0.001$). Here, the average number of participants from countries with a very high HDI is significantly higher than that of countries with a lower HDI level. For this category, the significant differences are evident not only in the overall period, but also in the individual decades and to a particular extent for countries with a low HDI.

Regarding the number of placements, significant differences between countries with a very high HDI and countries with a lower HDI are also evident in the overall period ($F(4,3.211) = 82.90$, $p < 0.001$). In both the overall period and the individual decades, countries with a very high HDI have a higher average number of placements than countries with a lower HDI.

Finally, when comparing the individual decades with each other, the average differences in the number of jury members, participants and placements between countries with a very high HDI and countries with a lower HDI decrease over time, but still represent significant differences across all decades. Finally, countries with a very high HDI dominate not only in jury membership and participant numbers, but also in placements.

A differentiation according to the self-portrayal and external portrayal of continents is shown in Table 23. This reveals how often a continent portrays itself in the individual decades, i.e., the continent from which the photographer comes and the continent where the photo was taken are identical, and how often a continent portrays other continents, i.e., the continent from which the photographer comes and the continent where the photo was taken are not identical.

**Table 23.** Over- and under-representation in the portrayal of the continents by themselves and by others.

| Period | Presentation | Africa | Asia | Australia/ Oceania | Europe | North America | South America | N |
|---|---|---|---|---|---|---|---|---|
| 1960–1969 | **Self** | 34 (2.5) | 96 (7.1) | 14 (1.0) | **955 (70.4)** | **223 (16.4)** | 35 (2.6) | 1357 (100) |
| | **External** | 5 (1.9) | 3 (1.1) | 6 (2.3) | **196 (74.2)** | **54 (20.5)** | - | 264 (100) |
| 1970–1979 | **Self** | 145 (7.4) | 75 (3.8) | 61 (3.1) | **1310 (67.2)** | **323 (16.6)** | 35 (1.8) | 1949 (100) |
| | **External** | 5 (1.0) | 4 (0.8) | 3 (0.6) | **340 (67.9)** | **147 (29.3)** | 2 (0.4) | 501 (100) |
| 1980–1989 | **Self** | 29 (1.8) | 84 (5.3) | 17 (1.1) | **811 (51.0)** | **633 (39.8)** | 17 (1.1) | 1591 (100) |
| | **External** | - | 6 (0.7) | 7 (0.8) | **425 (51.5)** | **380 (46.1)** | 7 (0.8) | 825 (100) |
| 1990–1999 | **Self** | 64 (6.2) | 53 (5.2) | 25 (2.4) | **569 (55.5)** | **291 (28.4)** | 23 (2.2) | 1025 (100) |
| | **External** | 14 (1.6) | 13 (1.5) | 46 (5.2) | **472 (53.5)** | **326 (36.9)** | 12 (1.4) | 883 (100) |
| 2000–2009 | **Self** | 59 (6.3) | 116 (12.4) | 72 (7.7) | **409 (43.8)** | **229 (24.5)** | 48 (5.1) | 933 (100) |
| | **External** | 14 (1.7) | 24 (2.9) | 36 (4.4) | **533 (64.4)** | **204 (24.7)** | 16 (1.9) | 827 (100) |
| 2010–2019 | **Self** | 44 (5.4) | 206 (25.3) | 15 (1.8) | **277 (34.0)** | **186 (22.8)** | 87 (10.7) | 815 (100) |
| | **External** | 19 (2.3) | 20 (2.5) | 62 (7.6) | **542 (66.8)** | **143 (17.6)** | 25 (3.1) | 811 (100) |
| Total Period | **Self** | 375 (4.9) | 630 (8.2) | 204 (2.7) | **4331 (56.5)** | **1885 (24.6)** | 245 (3.2) | 7670 (100) |
| | **External** | 57 (1.4) | 70 (1.7) | 160 (3.9) | **2508 (61.0)** | **1254 (30.5)** | 62 (1.5) | 4111 (100) |

Note: Relative frequencies by row; bold represents significant values.

As can be seen from the marginal frequencies, the proportion of self-representations has decreased over time, while the proportion of external representations has increased in every single decade. While in the period from 1960 to 1969 1357 images are self-representations and only 264 images are third party representations, in the period from 2010 to 2019 there are around 800 images in either group.

The number of self-representations of Europe and North America has been decreasing since the period from 1980 to 1989. For Europe in particular, third party representation has increased to such an extent that it dominates from the sixth decade (2000–2009) onwards. For North America, an increase in external depictions of the continent is also evident, with roughly equal numbers of both self- and third party depictions of North America from the 1990s onwards.

Over the period as a whole, Europe and North America are the continents with both the most frequent self-portrayals and portrayals of others. On the one hand, 56.5% of all images in which a continent depicts itself come from Europe and 24.6% of all such images come from North America. On the other hand, in the depiction of other continents Europe also dominates, with 61.0% of all depictions of others taken by European photographers, followed by 30.5% taken by photographers from North America.

In contrast to Europe and North America, Africa, Asia, Australia/Oceania and South America are primarily characterised by self-portrayals and there are significantly fewer cases in which photographers from these continents depict another continent. Of all shots in which a continent portrays itself, only small proportions are from Africa (4.9%), Asia (8.2%), Australia/Oceania (2.7%), or South America (3.2%).

Based on the high proportion of self-representations from Europe and North America, a differentiated view on how often these continents portray the others (with predominantly developing countries) follows in Table 24.

**Table 24.** External portrayal of Africa, Asia and South America by Europe and North America.

| Period | Presentation | Africa | Asia | South America | N |
|---|---|---|---|---|---|
| 1960–1969 | **Europe** | **79 (49.7)** | 66 (41.5) | 14 (8.8) | 159 (100) |
| | North America | 0 (0.0) | **42 (97.7)** | 1 (2.3) | 43 (100) |
| 1970–1979 | **Europe** | 66 (24.9) | **185 (69.8)** | 14(5.3) | 265 (100) |
| | North America | 29 (23.6) | **86 (69.9)** | 8 (6.5) | 123 (100) |
| 1980–1989 | **Europe** | 85 (25.1) | **223 (66.0)** | 30 (8.9) | 338 (100) |
| | North America | 51 (18.1) | **201 (71.3)** | 30 (10.6) | 282 (100) |
| 1990–1999 | **Europe** | **149 (40.7)** | 181 (49.5) | 36 (9.8) | 366 (100) |
| | North America | 74 (41.6) | **100 (56.2)** | 4 (2.2) | 178 (100) |
| 2000–2009 | **Europa** | **189 (40.4)** | 252 (53.8) | 27 (5.8) | 468 (100) |
| | North America | 27 (17.3) | **123 (78.8)** | 6 (3.9) | 156 (100) |
| 2010–2019 | **Europe** | **163 (35.2)** | 180 (38.9) | 120 (25.9) | 463 (100) |
| | North America | 50 (42.0) | **59 (49.6)** | 10 (8.4) | 119 (100) |
| Total Period | **Europe** | 731 (35.5) | **1087 (52.8)** | 241 (11.7) | **2059 (100)** |
| | North America | 231 (25.6) | **611 (67.8)** | 59 (6.6) | **901 (100)** |

Note: Relative frequencies by row; bold represents significant values.

As can be seen, photographers from Europe and North America most frequently portray Asia (52.8%, 67.8%), followed by Africa (35.5%, 25.6%). South America, on the other hand, is portrayed less frequently (Europe: 11.7%, North America: 6.6%). In total, Europe produces more than twice as many images of other continents as North America.

When comparing the self-portrayals and external portrayals of Tables 23 and 24, it is noticeable that Africa, Asia, and South America are all more often portrayed by photographers from Europe or North America than these continents portray themselves. While Africa portrays itself in 375 photos, a total of 962 images taken in Africa are by

photographers from Europe or North America. This means that only around one in four photos of Africa was taken by a photographer living in Africa. A similar ratio can be seen with regard to photos taken in Asia. While photographers from Asia portray their own continent in 630 cases, a total of 1698 photos were taken in Asia by photographers from Europe or North America. A smaller difference in the proportions of self-portrayals and portrayals by others can be seen for South America. In total, this continent depicted itself 245 times and was portrayed 300 times by European or North American photographers, so just under half of all photographs taken in South America were taken by photographers based there. The analysis of the forms of representation of the individual continents concludes with the addition of the HDI of the location where the photograph was taken.

Table 25 shows there are significant differences between the self- and external representations and the average HDI levels of the recording site across the overall period. First, the average HDI of a continent representing itself (M = 0.79, SD = 0.12) is higher than that of a continent representing others (M = 0.62, SD = 0.17). As shown by a t-test for independent samples, this difference is highly significant (t(3.862) = 39.12, $p < 0.001$). Thus, continents that represent themselves have on average a high HDI over the entire period, while continents represented by other continents have on average a medium HDI. This significant difference is visible in all three decades shown.

**Table 25.** Average HDI for the portrayal of a continent/country by themselves and by others.

| Period | Category | Identical | Mean HDI | SD | T | df | Significance | N |
|---|---|---|---|---|---|---|---|---|
| 1990–1999 | Continent | yes | **0.77** | 0.12 | 23.76 | 1148 | <0.001 | 945 |
| | | no | 0.58 | 0.17 | | | | 683 |
| | Country | yes | **0.80** | 0.10 | 27.55 | 1.625 | <0.001 | 636 |
| | | no | 0.62 | 0.16 | | | | 992 |
| 2000–2009 | Continent | yes | **0.80** | 0.13 | 28.20 | 1.348 | <0.001 | 888 |
| | | no | 0.58 | 0.18 | | | | 760 |
| | Country | yes | **0.82** | 0.11 | 28.15 | 1.639 | <0.001 | 655 |
| | | no | 0.62 | 0.18 | | | | 993 |
| 2010–2019 | Continent | yes | **0.81** | 0.12 | 17.95 | 1.426 | <0.001 | 809 |
| | | no | 0.68 | 0.15 | | | | 745 |
| | Country | yes | **0.81** | 0.12 | 15.63 | 1.542 | <0.001 | 617 |
| | | no | 0.70 | 0.16 | | | | 937 |
| Total Period | Continent | yes | **0.79** | 0.12 | 39.12 | 3.862 | <0.001 | 2642 |
| | | no | 0.62 | 0.17 | | | | 2188 |
| | Country | yes | **0.81** | 0.11 | 40.73 | 4.823 | <0.001 | 1908 |
| | | no | 0.65 | 0.17 | | | | 2922 |

Note: bold represents significant values.

Furthermore, significant differences in the average level of the HDI of the recording site are also evident with respect to the presentation forms of the countries. Across the overall period, the average HDI of a country that represents itself (M = 0.81, SD = 0.11) is higher than that of a country that is represented externally (M = 0.65, SD = 0.17). As shown by a t-test for independent variables, this difference is highly significant (t(4.823) = 40.73, $p < 0.001$). Accordingly, countries that portray themselves have on average a very high HDI over the entire period, while countries that are portrayed by other countries have on average a medium HDI. These distinctions are found in all decades and are each characterised by highly significant differences.

Interim Conclusion III

In analysing existing power structures in the World Press Photo competition, the initial focus was on the correlation between continents and number of jury members, participants, and placements. It emerges that two-thirds of all jury members and participants in the history of the competition have been from Europe. In addition, Europe occupied more than half of all placements ever given. Furthermore, North America's success is also evident in the high number of jury members from that continent and the number of placements it has taken. Europe's dominant role in the competition becomes particularly clear when the statistical analysis is taken into account. As the results make clear, there are significant differences between the average number of jury members, participants and placements and their origin (continent) in the overall period and in almost all categories of the individual decades. In this context, countries from Europe participating in the competition have a significantly higher average number of jury members, participants and placements than countries from Africa, Asia and South America participating in the competition.

Due to the over-representation of Europe in the competition, the results of the data analysis initially parallel those of König (2010), who identified a Eurocentric view for footage from Africa. The strong prominence of Europe is evident not only in the number of jury members, participants, and placements, but also in Europe's representation of Africa as well as Asia and South America. Furthermore, the results tie in with Godulla's (2009) study, which revealed a strong correlation between shots of crisis areas and the origin of the photographers among the winning photos. As the results show, Africa, Asia, and South America are more dominated by negative portrayals and especially images of conflict compared to Europe. At the same time, Europe dominates in the representation of these continents.

Finally, Hypothesis H7, which proposes that Europe is significantly over-represented in the number of jury members, participants, and placements compared to continents with predominantly developing countries, is accepted.

In a next step, taking the HDI into account, it is shown that there are significant differences between the average number of jury members, participants, and placements and the HDI level of the participating countries, both in the overall period and in the individual decades. Countries with a very high HDI participating in the competition have a higher average number of jury members, participants, and placements than countries with a lower HDI level (high, medium or low HDI), with the largest difference compared to countries with a low HDI.

Again, these results tie in with Godulla's (2009) findings that winning photos often come from photographers from Western countries and continents with predominantly industrialised nations.

Following on from this, an analysis of the HDI in the context of self-portrayal and portrayal by others of the continents and countries reveals that continents and countries that portray themselves have, on average, a high or very high HDI in the overall period and in the individual decades, while continents and countries that are portrayed by others have on average a medium HDI. As can be seen, the differences in the possibility of self-representation and third-party representation are not only visible for the continents, but also with regard to the HDI of a continent or country. Therefore, Hypothesis H8 is accepted in its entirety: The higher a country's HDI, the greater its success in terms of the number of judges, participants, and placements, as well as in terms of how it presents itself and how it is portrayed by others.

In a final step, the focus was placed on the differentiated consideration of self-portrayal and portrayal by others. It can be observed both in the overall period and in the individual decades that Europe primarily, but also North America, are the continents with the most frequent self-portrayals and portrayals of others. While Europe and North America have a very high proportion of images in which they portray other continents, Africa, Asia and South America predominantly portray themselves, with only a few portrayals of others.

As can be seen, the continent most frequently portrayed by Europe and North America is Asia, followed by Africa and South America. Overall, Europe dominates over North America in the representation of other continents. The high proportion of images in which Europe and North America depict the continents of Africa, Asia and South America outweighs these continents' own self-portrayals. Accordingly, just one in four photos of Africa or Asia and one in two photos of South America is by a photographer based there. Finally, it can be summarised that European and North American views and portrayals of continents with predominantly developing countries predominate and that these rather "cut-off" continents have significantly fewer opportunities to portray themselves.

The analysis of the photos from the yearbooks of the World Press Photo competition ties in with the results of the analysis of the winning photos, revealing a high proportion of photos from crisis regions (especially from Africa and Asia) and at the same time a high proportion of photographers from continents with predominantly industrialised nations. Prevailing power relations and imbalances in the possibility of self-portrayal and portrayal by others between the individual continents are also evident in the photographs in the yearbooks. Consequently, Hypothesis H9 is accepted: Europe and North America dominate when portraying other continents typified by developing countries (Africa, Asia, South America).

## 8. Discussion and Limitations

Obviously, any content analysis—even one as complex as this—is subject to certain limitations. In addition to the nature of the World Press Photos of the Year itself, the content analysis has succeeded in revealing significant distortions in the overall discourse. For example, it has become clear that the part of the world which can be characterised as "Western" and "highly developed" is far less often associated with conflict and catastrophe than the rest of the world. Negativism increases throughout the competition and particularly affects the continents of Africa, Asia, and South America. The more developed a country is, the less often it is connoted with negativism. Europe and North America have dominated the competition on many levels up to the present day. World Press Photo is in many ways a look at the rest of the world from the perspective of these two continents.

Although the question of transferability to the overall system of photojournalism cannot be answered based on these data, there are strong indications that World Press Photo is indeed the hoped-for distillation of a global phenomenon that would otherwise hardly be reducible to an analysable size. All protagonists—be they jury members, participants, or winners—have a professional background in photojournalism. Implicitly, they follow its institutionalised routines and are guided by the same selection mechanisms that can be assumed in editorial journalism.

What the content analyses cannot make visible, however, is the selection process in the competition itself. In principle, an essential level of analysis is currently lacking. Far fewer than one percent of the submitted photos end up being considered at all on the website, in the exhibition and in the yearbook. Conversely, this means that no empirically based statements can currently be made about the rejected 99 percent of the images. It would be highly relevant to investigate whether World Press Photo is merely reproducing a bias already present in the material or whether this bias is only produced and reinforced by the selection process.

This question can only be answered by a comprehensive input-output analysis, which at the same time seems suitable to provide valuable impulses for the discourse around image-based news value theory. In this context, it would also make sense to conduct qualitative guided interviews with jury members as a supplement. Participant observations would furthermore be beneficial to map the discussions taking place and to gain a better impression of the selection process. Many categories valuable to these studies can be derived from the present paper. It would be particularly desirable to further differentiate the content of the continuous increase of photos in the subject area of social documentation.

What had to be disregarded here due to the already high scope and complexity is the Digital Storytelling Contest organised by World Press Photo since 2011. In this specific contest, photos are integrated into a larger narrative context and linked with multimedia elements. The presentation thus leaves the context of the individual photo and expands into more complex overall narratives. Since the standard of specific excellence is also applied here, the approaches mentioned above could also generate valuable impulses for the further development of journalism research. In the case of both materials—individual photos and multimedia stories—it would furthermore be valuable to use reception studies to determine the effect of the material on the audience. This only makes sense, however, in the current time horizon, since, for example, the audience of the 1970s can no longer be observed or surveyed in the present for obvious reasons.

For image-based journalism research, the present study is able to show that digitisation has apparently not led to a broad democratisation of professional photojournalism—at least so far. Although photos can be taken in one part of the world and sent to another in a matter of seconds, the traditional dominance of Europe and North America apparently continues to the present day. In this respect, it is worthwhile to continue using World Press Photo as an object of study in the future.

**Author Contributions:** Conceptualization, A.G.; methodology, A.G., D.S. and R.P.; software, D.S.; validation, A.G., D.S. and R.P.; formal analysis, D.S., A.G. and R.P.; investigation, R.P., A.G. and D.S.; resources, A.G.; data curation, D.S., A.G. and R.P.; writing—original draft preparation, R.P., D.S. and A.G.; writing—review and editing, R.P., D.S. and A.G.; visualization, A.G. and D.S.; supervision, A.G.; project administration, A.G. All authors have read and agreed to the published version of the manuscript.

**Funding:** This research received no external funding.

**Institutional Review Board Statement:** Not applicable.

**Informed Consent Statement:** Not applicable.

**Data Availability Statement:** No publicly archived data set.

**Conflicts of Interest:** The authors declare no conflict of interest.

## Notes

[1] The number and names of the categories vary over the decades.

[2] In this overview, all year references in the photos refer to the year in which the photo was taken. For example, the World Press Photo 2020 was taken in 2019.

[3] Note: The years in the table header (1955–2020) refer to the years in which the World Press Photos of the Year were awarded, while the years within the table refer to the years in which the photos were taken.

[4] Note: Due to the lack of documentation of the number of participants in 2019, no information is available for this year on the frequency and origin of the persons participating in the competition. Accordingly, a slightly higher total number of participants can be assumed for the period from 2010 to 2019.

[5] Jury members for this decade were only found in internal documents made available by the foundation. There is no information concerning participants and rankings.

[6] This information was extracted from internal documents made available by the foundation, as well as from year books. No further information was available.

[5] Jury members for this decade were only found in internal documents made available by the foundation. There is no information concerning participants and rankings.

[6] This information was extracted from internal documents made available by the foundation, as well as from year books. No further information was available.

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
