# Peer review of "Whose Pictures, Whose Reality? Lines of Tradition in the Development of Topics, Negativity, and Power in the Photojournalistic Competition World Press Photo"

_journalmedia, doi:10.3390/journalmedia2040045_

Round 1

Reviewer 1 Report

This study is an extensive and complex undertaking. It covers a lot of ground. As a result it is dense in its description and findings. You might consider shortening it or splitting it into two or even three separate articles. The intro could be tightened up, as well. Other than that, it is well conceived, structured and well written. 

Reviewer 2 Report

Thank you for your manuscript. I have read it with great interest and found it fascinating.

The study examines all prize-winning photographs from the World Press Photo annual contests, between 1960 and 2020, in relation to topics, degree of negativity, and underlying power dynamics. The manuscript is very ambitious and makes a needed contribution to current discussions in photojournalism research field. The mixed-methods approach and research design employed are very complex, but consistent. The topics are timely, relevant an in need of both conceptual and empirical research.

Here are some comments/ suggestions for the author(s) to consider.

  • The manuscript as it stands seems to be an assemblage of different extracts of a book. The long-form in a supposedly article format makes it sometimes difficult to read. For instance, on page 13, it is written, “The research questions mentioned above will be answered in chapter 5 through a qualitative analysis of all World Press Photos of the Year awarded between 1955 and 2020.” As the journal publishes long-form articles, I would advise the author(s) to adapt the manuscript to an article format, removing all references to supposedly chapters, e.g., see chapter 5 or see chapter 6 on page 14 and so forth;
  • The literature review covers quite a few perspectives, but as it keeps unfolding it makes it difficult to remember what the article is about. I would advise the author(s) to do a critical reading through the manuscript identifying the main theoretical perspectives that can be relevant to their main line of argument. This would strengthen the author(s)’ argument and improve the clarity of the manuscript;
  • Coherence concerning the timeframe of the study across its parts still needs some work. The author(s) mentioned different timespans between and across the manuscript, such as 1960 and 2020, 1955 and 2020, and 1955 and 2019. For instance, on page 13, it is written, “The research questions mentioned above will be answered in chapter 5 through a qualitative analysis of all World Press Photos of the Year awarded between 1955 and 2020.” The statement contradicts the abstract where the author(s) state that the study examines all photos… between 1960 and 2020, and on page 15, in section 5, “Qualitative content analyses of all World Press Photos of the Year (1955-2019).”

Reviewer 3 Report

This is an ambitious study that assesses depictions across multiple decades of the World Press Photo competition. The author has been thorough in the description of results of the study. I commend the author for not only looking at the top-selected photograph from each year but then expanding to look at all the entries selected to be part of each year’s collection. Processing more than 11,000 photographs is a significant undertaking.

Many of the conclusions reached are not surprising. Europe and North America are portrayed more positively than other regions, and Europe and North America dominate representation in the content and judging of the competition. Several studies exist, some of them cited in this manuscript, that would suggest photographers working for organizations on those regions would focus on, or be assigned to cover, other regions in times of conflict, disaster, and so on. But without empirical study we can’t know for sure that is the case. In that sense, this study contributes to our understanding of one aspect of photojournalism: competitions and their perceived influence on the profession.

Before publication, some areas of the manuscript could be clarified, and additional support can be provided in some other areas.

On page 2, the author sets out to establish a “theoretical basis for this project.” I don’t see introduction and application of theory in this section. There are several concepts related to photojournalism listed. Is the author attempting to build theory from these concepts? If that’s the case, it should be presented and explained in the conclusion section. As it is written, it appears the author is identifying several characteristics or concepts related to photojournalism as a means of suggesting why the photographs reflect the ideas observed. If these are to be considered theory that can be used to predict anticipated outcomes, and is driving the analysis in this study, it needs to be more clearly presented as such.

Further explanation is also desired in the qualitative content analysis description. The author provides some description beginning on p. 15. The rationale for a qualitative analysis is clear and appropriate. In section 5.1 the author identifies two general methods, Q-sort and card sorting, but then leaves it up to the reader to know how those forms would dictate the process of the study. The authors should be more clear about the process of sorting the photographs and creating the titles and descriptions of categories that resulted. Did the authors follow the constant comparative method to sort the photographs? How was this process arrived at? How many coders were involved? Intercoder reliability is provided for the quantitative analysis. Were measures were taken to assess the categories and distribution in the qualitative analysis?

In section 5.2, was there an existing source for the categories of negativity identified? What is that source? Or do these categories come from the analysis of the photographs?

There is a difference in the number of categories observed in the qualitative analysis and the quantitative coding. The author indicates on p. 16 the six categories identified in the qualitative analysis can be used for the quantitative analysis and would be expanded on through theoretical considerations. On p. 23 the author lists 10 categories. Please provide a description of how the additional categories were added.

The author introduces categories of differentiation of negativity on p. 32, identifying abstract, concrete and massive levels of suffering, depictions of corpses, and so on. I did not notice a description of what defines these different categories. If I’m correct in that assessment, the author should identify what determines an abstract, concrete or massive representation.

I would add a few minor details:

On page 2, the author could add support to two points. First, on line45, the author begins to point out three photographs, referring to them as “the three most famous World Press Photos of the Year.” What is the measure by which it is determined they are the most famous? That is never explained.

On pabe 6 at line 244 the author states WPP “represents the biggest and most influential international competition for journalistic photography.” Can the author provide a citation to support this characterization? Several competitions claim to be influential, and international. Earlier in the manuscript a similar claim is attributed to WPP’s own characterization. Is that the source for this statement?

I’m unsure about a wording choice and the typical terminology for this journal. On pp. 13-14 the author refers to chapters 4 and 5, which would lead one to believe this manuscript was distilled from a larger work. I notice, however, that sections are numbered. Should the terminology be sections or chapters? Minor point, obviously.
